# Bayesian Context Aggregation for Neural Processes

**Michael Volpp**[1,2*]
**Fabian Flürenbrock**[1]
**Lukas Grossberger**[1]
**Christian Daniel**[1]
**Gerhard Neumann**[2,3]

[1]Bosch Center for Artificial Intelligence, Renningen, Germany
[2]Karlsruhe Institute of Technology, Karlsruhe, Germany
[3]University of Tübingen, Tübingen, Germany

## Abstract

Formulating scalable probabilistic regression models with reliable uncertainty estimates has been a long-standing challenge in machine learning research. Recently, casting probabilistic regression as a multi-task learning problem in terms of conditional latent variable (CLV) models such as the Neural Process (NP) has shown promising results. In this paper, we focus on context aggregation, a central component of such architectures, which fuses information from multiple context data points. So far, this aggregation operation has been treated separately from the inference of a latent representation of the target function in CLV models. Our key contribution is to combine these steps into one holistic mechanism by phrasing context aggregation as a Bayesian inference problem. The resulting Bayesian Aggregation (BA) mechanism enables principled handling of task ambiguity, which is key for efficiently processing context information. We demonstrate on a range of challenging experiments that BA consistently improves upon the performance of traditional mean aggregation while remaining computationally efficient and fully compatible with existing NP-based models.

## 1 Introduction

Estimating statistical relationships between physical quantities from measured data is of central importance in all branches of science and engineering and devising powerful regression models for this purpose forms a major field of study in statistics and machine learning. When judging representative power, neural networks (NNs) are arguably the most prominent member of the regression toolbox. NNs cope well with large amounts of training data and are computationally efficient at test time. On the downside, standard NN variants do not provide uncertainty estimates over their predictions and tend to overfit on small datasets. Gaussian processes (GPs) may be viewed as complementary to NNs as they provide reliable uncertainty estimates but their cubic (quadratic) scaling with the number of context data points at training (test) time in their basic formulation affects the application on tasks with large amounts of data or on high-dimensional problems.

Recently, a lot of interest in the scientific community is drawn to combinations of aspects of NNs and GPs. Indeed, a prominent formulation of probabilistic regression is as a multi-task learning problem formalized in terms of amortized inference in conditional latent variable (CLV) models, which results in NN-based architectures which learn a distribution over target functions. Notable variants are given by the Neural Process (NP) (Garnelo et al., 2018b) and the work of Gordon et al. (2019), which presents a unifying view on a range of related approaches in the language of CLV models.

Inspired by this research, we study context aggregation, a central component of such models, and propose a new, fully Bayesian, aggregation mechanism for CLV-based probabilistic regression models.

---

*Correspondence to: `Michael.Volpp@de.bosch.com`

To transform the information contained in the context data into a latent representation of the target function, current approaches typically employ a mean aggregator and feed the output of this aggregator into a NN to predict a distribution over global latent parameters of the function. Hence, aggregation and latent parameter inference have so far been treated as separate parts of the learning pipeline. Moreover, when using a mean aggregator, every context sample is assumed to carry the same amount of information. Yet, in practice, different input locations have different task ambiguity and, therefore, samples should be assigned different importance in the aggregation process. In contrast, our Bayesian aggregation mechanism treats context aggregation and latent parameter inference as one holistic mechanism, i.e., the aggregation directly yields the distribution over the latent parameters of the target function. Indeed, we formulate context aggregation as Bayesian inference of latent parameters using Gaussian conditioning in the latent space. Compared to existing methods, the resulting aggregator improves the handling of task ambiguity, as it can assign different variance levels to the context samples. This mechanism improves predictive performance, while it remains conceptually simple and introduces only negligible computational overhead. Moreover, our Bayesian aggregator can also be applied to deterministic model variants like the Conditional NP (CNP) (Garnelo et al., 2018a).

In summary, our contributions are (i) a novel Bayesian Aggregation (BA) mechanism for context aggregation in NP-based models for probabilistic regression, (ii) its application to existing CLV architectures as well as to deterministic variants like the CNP, and (iii) an exhaustive experimental evaluation, demonstrating BA's superiority over traditional mean aggregation.

## 2 RELATED WORK

Prominent approaches to probabilistic regression are Bayesian linear regression and its kernelized counterpart, the Gaussian process (GP) (Rasmussen and Williams, 2005). The formal correspondence of GPs with infinite-width Bayesian NNs (BNNs) has been established in Neal (1996) and Williams (1996). A broad range of research aims to overcome the cubic scaling behaviour of GPs with the number of context points, e.g., through sparse GP approximations (Smola and Bartlett, 2001; Lawrence et al., 2002; Snelson and Ghahramani, 2005; Quiñonero-Candela and Rasmussen, 2005), by deep kernel learning (Wilson et al., 2016), by approximating the posterior distribution of BNNs (MacKay, 1992; Hinton and van Camp, 1993; Gal and Ghahramani, 2016; Louizos and Welling, 2017), or, by adaptive Bayesian linear regression, i.e., by performing inference over the last layer of a NN which introduces sparsity through linear combinations of finitely many learned basis functions (Lazaro-Gredilla and Figueiras-Vidal, 2010; Hinton and Salakhutdinov, 2008; Snoek et al., 2012; Calandra et al., 2016). An in a sense complementary approach aims to increase the data-efficiency of deep architectures by a fully Bayesian treatment of hierarchical latent variable models ("DeepGPs") (Damianou and Lawrence, 2013).

A parallel line of research studies probabilistic regression in the multi-task setting. Here, the goal is to formulate models which are data-efficient on an unseen target task by training them on data from a set of related source tasks. Bardenet et al. (2013); Yogatama and Mann (2014), and Golovin et al. (2017) study multi-task formulations of GP-based models. More general approaches of this kind employ the meta-learning framework (Schmidhuber, 1987; Thrun and Pratt, 1998; Vilalta and Drissi, 2005), where a model's training procedure is formulated in a way which incentivizes it to learn *how* to solve unseen tasks rapidly with only a few context examples ("learning to learn", "few-shot learning" (Fei-Fei et al., 2006; Lake et al., 2011)). A range of such methods trains a meta-learner to learn how to adjust the parameters of the learner's model (Bengio et al., 1991; Schmidhuber, 1992), an approach which has recently been applied to few-shot image classification (Ravi and Larochelle, 2017), or to learning data-efficient optimization algorithms (Hochreiter et al., 2001; Li and Malik, 2016; Andrychowicz et al., 2016; Chen et al., 2017; Perrone et al., 2018; Volpp et al., 2019). Other branches of meta-learning research aim to learn similarity metrics to determine the relevance of context samples for the target task (Koch et al., 2015; Vinyals et al., 2016; Snell et al., 2017; Sung et al., 2017), or explore the application of memory-augmented neural networks for meta-learning (Santoro et al., 2016). Finn et al. (2017) propose model-agnostic meta-learning (MAML), a general framework for fast parameter adaptation in gradient-based learning methods.

A successful formulation of probabilistic regression as a few-shot learning problem in a multi-task setting is enabled by recent advances in the area of *probabilistic* meta-learning methods which allow a quantitative treatment of the uncertainty arising due to task ambiguity, a feature particularly

relevant for few-shot learning problems. One line of work specifically studies probabilistic extensions of MAML (Grant et al., 2018; Ravi and Larochelle, 2017; Rusu et al., 2018; Finn et al., 2018; Kim et al., 2018). Further important approaches are based on amortized inference in multi-task CLV models (Heskes, 2000; Bakker and Heskes, 2003; Kingma and Welling, 2013; Rezende et al., 2014; Sohn et al., 2015), which forms the basis of the Neural Statistician proposed by Edwards and Storkey (2017) and of the NP model family (Garnelo et al., 2018b; Kim et al., 2019; Louizos et al., 2019). Gordon et al. (2019) present a unifying view on many of the aforementioned probabilistic architectures. Building on the conditional NPs (CNPs) proposed by Garnelo et al. (2018a), a range of NP-based architectures, such as Garnelo et al. (2018b) and Kim et al. (2019), consider combinations of deterministic and CLV model architectures. Recently, Gordon et al. (2020) extended CNPs to include translation equivariance in the input space, yielding state-of-the-art predictive performance.

In this paper, we also employ a formulation of probabilistic regression in terms of a multi-task CLV model. However, while in previous work the context aggregation mechanism (Zaheer et al., 2017; Wagstaff et al., 2019) was merely viewed as a necessity to consume context sets of variable size, we take inspiration from Becker et al. (2019) and emphasize the fundamental connection of latent parameter inference with context aggregation and, hence, base our model on a novel Bayesian aggregation mechanism.

## 3 PRELIMINARIES

We present the standard multi-task CLV model which forms the basis for our discussion and present traditional mean context aggregation (MA) and the variational inference (VI) likelihood approximation as employed by the NP model family (Garnelo et al., 2018a; Kim et al., 2019), as well as an alternative Monte Carlo (MC)-based approximation.

**Problem Statement.** We frame probabilistic regression as a multi-task learning problem. Let $\mathcal{F}$ denote a family of functions $f_\ell : \mathbb{R}^{d_x} \to \mathbb{R}^{d_y}$ with some form of shared statistical structure. We assume to have available data sets $\mathcal{D}_\ell \equiv \{(x_{\ell,i}, y_{\ell,i})\}_i$ of evaluations $y_{\ell,i} \equiv f_\ell(x_{\ell,i}) + \varepsilon$ from a subset of functions ("tasks") $\{f_\ell\}_{\ell=1}^L \subset \mathcal{F}$ with additive Gaussian noise $\varepsilon \sim \mathcal{N}\left(0, \sigma_n^2\right)$. From this data, we aim to learn the posterior predictive distribution $p\left(y_\ell \middle| x_\ell, \mathcal{D}_\ell^c\right)$ over a (set of) $y_\ell$, given the corresponding (set of) inputs $x_\ell$ as well as a context set $\mathcal{D}_\ell^c \subset \mathcal{D}_\ell$.

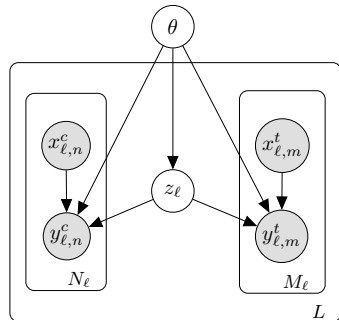

**The Multi-Task CLV Model.** We formalize the multi-task learning problem in terms of a CLV model (Heskes, 2000; Gordon et al., 2019) as shown in Fig. 1. The model employs task-specific global latent variables $z_\ell \in \mathbb{R}^{d_z}$, as well as a task-independent latent variable $\theta$, capturing the statistical structure shared between tasks. To learn $\theta$, we split the data into context sets $\mathcal{D}_\ell^c \equiv \{(x_{\ell,n}^c, y_{\ell,n}^c)\}_{n=1}^{N_\ell}$ and target sets $\mathcal{D}_\ell^t \equiv \{(x_{\ell,m}^t, y_{\ell,m}^t)\}_{m=1}^{M_\ell}$ and maximize the posterior predictive likelihood function

Figure 1: Multi-task CLV model with task-specific global latent variables $z_\ell$ and a task-independent variable $\theta$ describing statistical structure shared between tasks.

$$\prod_{\ell=1}^L p\left(y_{\ell,1:M_\ell}^t \middle| x_{\ell,1:M_\ell}^t, \mathcal{D}_\ell^c, \theta\right) = \prod_{\ell=1}^L \int p(z_\ell \mid \mathcal{D}_\ell^c, \theta) \prod_{m=1}^{M_\ell} p\left(y_{\ell,m}^t \middle| z_\ell, x_{\ell,m}^t, \theta\right) \mathrm{d}z_\ell \quad (1)$$

w.r.t. $\theta$. In what follows, we omit task indices $\ell$ to avoid clutter.

**Likelihood Approximation.** Marginalizing over the task-specific latent variables $z$ is intractable for reasonably complex models, so one has to employ some form of approximation. The NP-family of models (Garnelo et al., 2018b; Kim et al., 2019) uses an approximation of the form

$$\log p\left(y_{1:M}^t \middle| x_{1:M}^t, \mathcal{D}^c, \theta\right) \gtrapprox \mathbb{E}_{q_\phi(z\mid \mathcal{D}^c \cup \mathcal{D}^t)}\left[\sum_{m=1}^M \log p\left(y_m^t \middle| z, x_m^t, \theta\right) + \log \frac{q_\phi\left(z \mid \mathcal{D}^c\right)}{q_\phi\left(z \mid \mathcal{D}^c \cup \mathcal{D}^t\right)}\right].$$

$$(2)$$

Being derived using a variational approach, this approximation utilizes an approximate posterior distribution $q_\phi\left(z|\mathcal{D}^c\right) \approx p\left(z|\mathcal{D}^c, \theta\right)$. Note, however, that it does not constitute a proper evidence lower bound for the posterior predictive likelihood since the intractable latent posterior $p\left(z|\mathcal{D}^c, \theta\right)$ has been replaced by $q_\phi\left(z|\mathcal{D}^c\right)$ in the nominator of the rightmost term (Le et al., 2018). An alternative approximation, employed for instance in Gordon et al. (2019), also replaces the intractable latent posterior distribution by an approximate distribution $q_\phi\left(z|\mathcal{D}^c\right) \approx p\left(z|\mathcal{D}^c, \theta\right)$ and uses a Monte-Carlo (MC) approximation of the resulting integral based on $K$ latent samples, i.e.,

$$\log p\left(y_{1:M}^t \middle| x_{1:M}^t, \mathcal{D}^c, \theta\right) \approx -\log K + \log \sum_{k=1}^{K} \prod_{m=1}^{M} p\left(y_m^t \middle| z_k, x_m^t, \theta\right), \quad z_k \sim q_\phi\left(z|\mathcal{D}^c\right). \quad (3)$$

Note that both approaches employ approximations $q_\phi\left(z|\mathcal{D}^c\right)$ of the latent posterior distribution $p\left(z|\mathcal{D}^c, \theta\right)$ and, as indicated by the notation, amortize inference in the sense that one single set of parameters $\phi$ is shared between all context data points. This enables efficient inference at test time, as no per-data-point optimization loops are required. As is standard in the literature (Garnelo et al., 2018b; Kim et al., 2019), we represent $q_\phi\left(z|\mathcal{D}^c\right)$ and $p\left(y_m^t|z, x_m^t, \theta\right)$ by NNs and refer to them as the encoder (enc, parameters $\phi$) and decoder (dec, parameters $\theta$) networks, respectively. These networks set the means and variances of factorized Gaussian distributions, i.e.,

$$q_\phi\left(z|\mathcal{D}^c\right) = \mathcal{N}\left(z|\mu_z, \mathrm{diag}\left(\sigma_z^2\right)\right), \quad \mu_z = \mathrm{enc}_{\mu_z,\phi}\left(\mathcal{D}^c\right), \quad \sigma_z^2 = \mathrm{enc}_{\sigma_z^2,\phi}\left(\mathcal{D}^c\right), \quad (4)$$

$$p\left(y_m^t \middle| z, x_m^t, \theta\right) = \mathcal{N}\left(y_m^t \middle| \mu_y, \mathrm{diag}\left(\sigma_y^2\right)\right), \quad \mu_y = \mathrm{dec}_{\mu_y,\theta}\left(z, x_m^t\right), \quad \sigma_y^2 = \mathrm{dec}_{\sigma_y^2,\theta}\left(z, x_m^t\right). \quad (5)$$

**Context Aggregation.** The latent variable $z$ is global in the sense that it depends on the whole context set $\mathcal{D}^c$. Therefore, some form of aggregation mechanism is required to enable the encoder to consume context sets $\mathcal{D}^c$ of variable size. To represent a meaningful operation on sets, such an aggregation mechanism has to be invariant to permutations of the context data points. Zaheer et al. (2017) characterize possible aggregation mechanisms w.r.t. this permutation invariance condition, resulting in the structure of traditional aggregation mechanisms depicted in Fig. 2(a). Each context data tuple $(x_n^c, y_n^c)$ is first mapped onto a latent observation $r_n = \mathrm{enc}_{r,\phi}\left(x_n^c, y_n^c\right) \in \mathbb{R}^{d_r}$. Then, a permutation-invariant operation is applied to the set $\{r_n\}_{n=1}^N$ to obtain an aggregated latent observation $\bar{r}$. One prominent choice, employed for instance in Garnelo et al. (2018a), Kim et al. (2019), and Gordon et al. (2019), is to take the mean, i.e.,

$$\bar{r} = \frac{1}{N} \sum_{n=1}^{N} r_n. \quad (6)$$

Subsequently, $\bar{r}$ is mapped onto the parameters $\mu_z$ and $\sigma_z^2$ of the approximate posterior distribution $q_\phi\left(z|\mathcal{D}^c\right)$ using additional encoder networks, i.e., $\mu_z = \mathrm{enc}_{\mu_z,\phi}\left(\bar{r}\right)$ and $\sigma_z^2 = \mathrm{enc}_{\sigma_z^2,\phi}\left(\bar{r}\right)$. Note that three encoder networks are employed here: (i) $\mathrm{enc}_{r,\phi}$ to map from the context pairs to $r_n$, (ii) $\mathrm{enc}_{\mu_z,\phi}$ to compute $\mu_z$ from the aggregated mean $\bar{r}$ and (iii) $\mathrm{enc}_{\sigma_z^2,\phi}$ to compute the variance $\sigma_z^2$ from $\bar{r}$. In what follows, we refer to this aggregation mechanism as mean aggregation (MA) and to the networks $\mathrm{enc}_{\mu_z,\phi}$ and $\mathrm{enc}_{\sigma_z^2,\phi}$ collectively as "$\bar{r}$-to-$z$-networks".

## 4  BAYESIAN CONTEXT AGGREGATION

We propose Bayesian Aggregation (BA), a novel context data aggregation technique for CLV models which avoids the detour via an aggregated latent observation $\bar{r}$ and directly treats the object of interest, namely the latent variable $z$, as the aggregated quantity. This reflects a central observation for CLV models with global latent variables: *context data aggregation and hidden parameter inference are fundamentally the same mechanism*. Our key insight is to define a probabilistic observation model $p(r|z)$ for $r$ which depends on $z$. Given a new latent observation $r_n = \mathrm{enc}_{r,\phi}(x_n^c, y_n^c)$, we can update $p(z)$ by computing the posterior $p(z|r_n) = p(r_n|z)p(z)/p(r_n)$. Hence, by formulating context data aggregation as a Bayesian inference problem, we aggregate the information contained in $\mathcal{D}^c$ directly into the statistical description of $z$ based on first principles.

### 4.1  BAYESIAN CONTEXT AGGREGATION VIA GAUSSIAN CONDITIONING

BA can easily be implemented using a factorized Gaussian observation model of the form

$$p\left(r_n|z\right) = \mathcal{N}\left(r_n|z, \mathrm{diag}(\sigma_{r_n}^2)\right), \quad r_n = \mathrm{enc}_{r,\phi}\left(x_n^c, y_n^c\right), \quad \sigma_{r_n}^2 = \mathrm{enc}_{\sigma_r^2,\phi}\left(x_n^c, y_n^c\right). \quad (7)$$

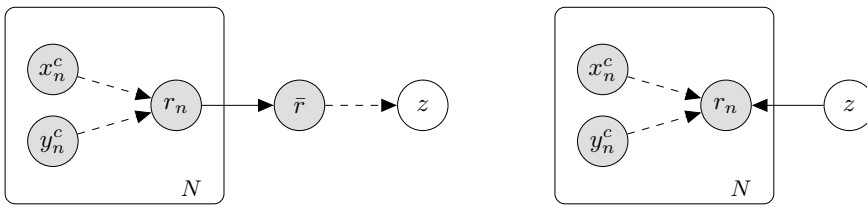

(a) Traditional mean aggregation (MA).          (b) Our Bayesian aggregation (BA).

Figure 2: Comparison of aggregation mechanisms in CLV models. Dashed lines correspond to learned components of the posterior approximation $q_\phi\left(z\,|\,\mathcal{D}^c\right)$. BA avoids the detour via a mean-aggregated latent observation $\bar{r}$ and aggregates $\mathcal{D}^c$ directly in the statistical description of $z$. This allows to incorporate a quantification of the information content of each context tuple $(x_n^c, y_n^c)$ as well as of $z$ into the inference in a principled manner, while MA assigns the same weight to each context tuple.

Note that, in contrast to standard variational auto-encoders (VAEs) (Kingma and Welling, 2013), we do not learn the mean and variance of a Gaussian distribution, but we learn the latent observation $r_n$ (which can be considered as a sample of $p(z)$) together with the variance $\sigma_{r_n}^2$ of this observation. This architecture allows the application of Gaussian conditioning while this is difficult for VAEs. Indeed, we impose a factorized Gaussian prior $p_0\left(z\right) \equiv \mathcal{N}\left(z\,|\,\mu_{z,0}, \mathrm{diag}\left(\sigma_{z,0}^2\right)\right)$ and arrive at a Gaussian aggregation model which allows to derive the parameters of the posterior distribution $q_\phi\left(z\,|\,\mathcal{D}^c\right)$ in closed form[1] (cf. App. 7.1):

$$\sigma_z^2 = \left[\left(\sigma_{z,0}^2\right)^\ominus + \sum_{n=1}^{N}\left(\sigma_{r_n}^2\right)^\ominus\right]^\ominus, \quad \mu_z = \mu_{z,0} + \sigma_z^2 \odot \sum_{n=1}^{N}\left(r_n - \mu_{z,0}\right) \oslash \left(\sigma_{r_n}^2\right). \qquad (8)$$

Here $^\ominus$, $\odot$ and $\oslash$ denote element-wise inversion, product, and division, respectively. These equations naturally lend themselves to efficient incremental updates as new context data $\left(x_n^c, y_n^c\right)$ arrives by using the current posterior parameters $\mu_{z,\mathrm{old}}$ and $\sigma_{z,\mathrm{old}}^2$ in place of the prior parameters, i.e.,

$$\sigma_{z,\mathrm{new}}^2 = \left[\left(\sigma_{z,\mathrm{old}}^2\right)^\ominus + \left(\sigma_{r_n}^2\right)^\ominus\right]^\ominus, \quad \mu_z = \mu_{z,\mathrm{old}} + \sigma_{z,\mathrm{new}}^2 \odot \left(r_n - \mu_{z,\mathrm{old}}\right) \oslash \left(\sigma_{r_n}^2\right). \qquad (9)$$

BA employs two encoder networks, $\mathrm{enc}_{r,\phi}$ and $\mathrm{enc}_{\sigma_r^2,\phi}$, mapping context tuples to latent observations and their variances, respectively. In contrast to MA, it does not require $\bar{r}$-to-$z$-networks, because the set $\{r_n\}_{n=1}^{N}$ is aggregated directly into the statistical description of $z$ by means of Eq. (8), cf. Fig. 2(b). Note that our factorization assumptions avoid the expensive matrix inversions that typically occur in Gaussian conditioning and which are difficult to backpropagate. Using factorized distributions renders BA cheap to evaluate with only marginal computational overhead in comparison to MA. Furthermore, we can easily backpropagate through BA to compute gradients to optimize the parameters of the encoder and decoder networks. As the latent space $z$ is shaped by the encoder network, the factorization assumptions are valid because the network will find a space where these assumptions work well. Note further that BA represents a permutation-invariant operation on $\mathcal{D}^c$.

**Discussion.** BA includes MA as a special case. Indeed, Eq. (8) reduces to the mean-aggregated latent observation Eq. (6) if we impose a non-informative prior and uniform observation variances $\sigma_{r_n}^2 \equiv 1$.[2] This observation sheds light on the benefits of a Bayesian treatment of aggregation. MA assigns the same weight $1/N$ to each latent observation $r_n$, independent of the amount of information contained in the corresponding context data tuple $(x_n^c, y_n^c)$, as well as independent of the uncertainty about the current estimation of $z$. Bayesian aggregation remedies both of these limitations: the influence of $r_n$ on the parameters $\mu_{z,\mathrm{old}}$ and $\sigma_{z,\mathrm{old}}^2$ describing the current aggregated state is determined by the relative magnitude of the observation variance $\sigma_{r_n}^2$ and the latent variance

---

[1]Note that an extended observation model of the form $p\left(r_n\,|\,z\right) = \mathcal{N}\left(r_n\,|\,z + \mu_{r_n}, \mathrm{diag}(\sigma_{r_n}^2)\right)$, with $\mu_{r_n}$ given by a third encoder output, does not lead to a more expressive aggregation mechanism. Indeed, the resulting posterior variances would stay unchanged and the posterior mean would read $\mu_z = \mu_{z,0} + \sigma_z^2 \odot \sum_{n=1}^{N}\left(r_n - \mu_{r_n} - \mu_{z,0}\right) \oslash \left(\sigma_{r_n}^2\right)$. Therefore, we would just subtract two distinct encoder outputs computed from the same inputs, resulting in exactly the same expressivity, which is why we set $\mu_{r_n} \equiv 0$.

[2]As motivated above, we consider $\bar{r}$ as the aggregated quantity of MA and the distribution over $z$, described by $\mu_z$ and $\sigma_z^2$, as the aggregated quantity of BA. Note that Eq. (8) does not necessarily generalize $\mu_z$ and $\sigma_z^2$ after nonlinear $\bar{r}$-to-$z$-networks.

$\sigma_{z,\text{old}}^2$, cf. Eq. (9). This emphasizes the central role of the learned observation variances $\sigma_{r_n}^2$: they allow to quantify the amount of information contained in each latent observation $r_n$. BA can therefore handle task ambiguity more efficiently than MA, as the architecture can learn to assign little weight (by predicting high observation variances $\sigma_{r_n}^2$) to context points $(x_n^c, y_n^c)$ located in areas with high task ambiguity, i.e., to points which could have been generated by many of the functions in $\mathcal{F}$. Conversely, in areas with little task ambiguity, i.e., if $(x_n^c, y_n^c)$ contains a lot of information about the underlying function, BA can induce a strong influence on the posterior latent distribution. In contrast, MA has to find ways to propagate such information through the aggregation mechanism by encoding it in the mean-aggregated latent observation $\bar{r}$.

## 4.2 Likelihood Approximation with Bayesian Context Aggregation

We show that BA is versatile in the sense that it can replace traditional MA in various CLV-based NP architectures as proposed, e.g., in Garnelo et al. (2018b) and Gordon et al. (2019), which employ samples from the approximate latent posterior $q_\phi(z \mid \mathcal{D}^c)$ to approximate the likelihood (as discussed in Sec. 3), as well as in deterministic variants like the CNP (Garnelo et al., 2018a).

**Sampling-Based Likelihood Approximations.** BA is naturally compatible with both the VI and MC likelihood approximations for CLV models. Indeed, BA defines a Gaussian latent distribution from which we can easily obtain samples $z$ in order to evaluate Eq. (2) or Eq. (3) using the decoder parametrization Eq. (5).

**Bayesian Context Aggregation for Conditional Neural Processes.** BA motivates a novel, alternative, method to approximate the posterior predictive likelihood Eq. (1), resulting in a deterministic loss function which can be efficiently optimized for $\theta$ and $\phi$ in an end-to-end fashion. To this end, we employ a Gaussian approximation of the posterior predictive likelihood of the form

$$p\left(y_{1:M}^t \mid x_{1:M}^t, \mathcal{D}^c, \theta\right) \approx \mathcal{N}\left(y_{1:M}^t \mid \mu_y, \Sigma_y\right). \tag{10}$$

This is inspired by GPs which also define a Gaussian likelihood. Maximizing this expression yields the optimal solution $\mu_y = \tilde{\mu}_y$, $\Sigma_y = \tilde{\Sigma}_y$, with $\tilde{\mu}_y$ and $\tilde{\Sigma}_y$ being the first and second moments of the true posterior predictive distribution. This is a well-known result known as *moment matching*, a popular variant of deterministic approximate inference used, e.g., in Deisenroth and Rasmussen (2011) and Becker et al. (2019). $\tilde{\mu}_y$ and $\tilde{\Sigma}_y$ are functions of the moments $\mu_z$ and $\sigma_z^2$ of the latent posterior $p(z \mid \mathcal{D}^c, \theta)$ which motivates the following decoder parametrization:

$$\mu_y = \text{dec}_{\mu_y,\theta}\left(\mu_z, \sigma_z^2, x_m^t\right), \quad \sigma_y^2 = \text{dec}_{\sigma_y^2,\theta}\left(\mu_z, \sigma_z^2, x_m^t\right), \quad \Sigma_y = \text{diag}\left(\sigma_y^2\right). \tag{11}$$

Here, $\mu_z$ and $\sigma_z^2$ are given by the BA Eqs. (8). Note that we define the Gaussian approximation to be factorized w.r.t. individual $y_m^t$, an assumption which simplifies the architecture but could be dropped if a more expressive model was required. This decoder can be interpreted as a "moment matching network", computing the moments of $y$ given the moments of $z$. Indeed, in contrast to decoder networks of CLV-based NP architectures as defined in Eq. (5), it operates on the moments $\mu_z$ and $\sigma_z^2$ of the latent distribution instead of on samples $z$ which allows to evaluate this approximation in a deterministic manner. In this sense, the resulting model is akin to the CNP which defines a deterministic, conditional model with a decoder operating on the mean-aggregated latent observation $\bar{r}$. However, BA-based models trained in this deterministic manner still benefit from BA's ability to accurately quantify latent parameter uncertainty which yields significantly improved predictive likelihoods. In what follows, we refer to this approximation scheme as direct parameter-based (PB) likelihood optimization.

**Discussion.** The concrete choice of likelihood approximation or, equivalently, model architecture depends mainly on the intended use-case. Sampling-based models are generally more expressive as they can represent complex, i.e., structured, non-Gaussian, posterior predictive distributions. Moreover, they yield true function samples while deterministic models only allow approximate function samples through auto-regressive (AR) sampling schemes. Nevertheless, deterministic models exhibit several computational advantages. They yield direct probabilistic predictions in a single forward pass, while the predictions of sampling-based methods are only defined through averages over multiple function samples and hence require multiple forward passes. Likewise, evaluating the MC-based likelihood approximation Eq. (3) during training requires to draw multiple

Table 1: Posterior predictive log-likelihood on functions drawn from GP priors with RBF, weakly periodic, and Matern-5/2 kernels, averaged over context sets with $N \in \{0, 1, \ldots, 64\}$ points (table) and in dependence of $N$ (figure). BA consistently outperforms MA, independent of the likelihood approximation, with MC being the most expressive choice. PB represents an efficient, deterministic alternative, while the VI approximation tends to perform worst, in particular for small $N$.

| | PB/det. | | VI | | MC | | ANP |
| | BA | MA (CNP) | BA | MA (LP-NP) | BA | MA | MA + Attention |
|---|---|---|---|---|---|---|---|
| RBF GP | $\mathbf{1.37 \pm 0.15}$ | $0.94 \pm 0.04$ | $\mathbf{1.40 \pm 0.04}$ | $0.45 \pm 0.12$ | $\mathbf{1.62 \pm 0.05}$ | $1.07 \pm 0.05$ | $0.98 \pm 0.02$ |
| Weakly Periodic GP | $\mathbf{1.13 \pm 0.08}$ | $0.76 \pm 0.02$ | $\mathbf{0.89 \pm 0.03}$ | $0.07 \pm 0.14$ | $\mathbf{1.30 \pm 0.06}$ | $0.85 \pm 0.04$ | $1.02 \pm 0.02$ |
| Matern-5/2 GP | $\mathbf{-0.50 \pm 0.07}$ | $-0.68 \pm 0.01$ | $\mathbf{-0.79 \pm 0.01}$ | $-1.09 \pm 0.10$ | $\mathbf{-0.33 \pm 0.01}$ | $-0.90 \pm 0.15$ | $\mathbf{0.25 \pm 0.02}$ |

$(K)$ latent samples $z$. While the VI likelihood approximation Eq. (2) can be optimized on a single function sample per training step through stochastic gradient descent (Bishop, 2006), it has the disadvantage that it requires to feed target sets $\mathcal{D}^t$ through the encoder which can impede the training for small context sets $\mathcal{D}^c$ as discussed in detail in App. 7.2.

## 5 EXPERIMENTS

We present experiments to compare the performances of BA and of MA in NP-based models. To provide a complete picture, we evaluate all combinations of likelihood approximations (PB/deterministic Eq. (10), VI Eq. (2), MC Eq. (3)) and aggregation methods (BA Eq. (8), MA Eq. (6)), resulting in six different model architectures, cf. Fig. 4 in App. 7.5.2. Two of these architectures correspond to existing members of the NP family: MA + deterministic is equivalent to the CNP (Garnelo et al., 2018a), and MA + VI corresponds to the Latent-Path NP (LP-NP) (Garnelo et al., 2018b), i.e., the NP without a deterministic path. We further evaluate the Attentive Neural Process (ANP) (Kim et al., 2019), which employs a hybrid approach, combining LP-NP with a cross-attention mechanism in a parallel deterministic path[3], as well as an NP-architecture using MA with a self-attentive (SA) encoder network. Note that BA can also be used in hybrid models like ANP or in combination with SA, an idea we leave for future research. In App. 7.4 we discuss NP-based regression in relation to other methods for (scalable) probabilistic regression.

The performance of NP-based models depends heavily on the encoder and decoder network architectures as well as on the latent space dimensionality $d_z$. To assess the influence of the aggregation mechanism independently from all other confounding factors, we consistently optimize the encoder and decoder network architectures, the latent-space dimensionality $d_z$, as well as the learning rate of the Adam optimizer (Kingma and Ba, 2015), *independently for all model architectures and for all experiments* using the Optuna (Akiba et al., 2019) framework, cf. App. 7.5.3. If not stated differently, we report performance in terms of the mean posterior predictive log-likelihood over 256 test tasks with 256 data points each, conditioned on context sets containing $N \in \{0, 1, \ldots, N_{\max}\}$ data points (cf. App. 7.5.4). For sampling-based methods (VI, MC, ANP), we report the joint log-likelihood over the test sets using a Monte-Carlo approximation with 25 latent samples, cf. App. 7.5.4. We average the resulting log-likelihood values over 10 training runs with different random seeds and report 95% confidence intervals. We publish source code to reproduce the experimental results online.[4]

**GP Samples.** We evaluate the architectures on synthetic functions drawn from GP priors with different kernels (RBF, weakly periodic, Matern-5/2), as proposed by Gordon et al. (2020), cf. App. 7.5.1. We generate a new batch of functions for each training epoch. The results (Tab. 1) show that BA consistently outperforms MA, independent of the model architecture. In-

---

[3]For ANP, we use original code from `https://github.com/deepmind/neural-processes`
[4]`https://github.com/boschresearch/bayesian-context-aggregation`

Table 3: Posterior predictive log-likelihood on 1D and 3D quadratic functions with limited numbers $L$ of training tasks, averaged over context sets with $N \in \{0, 1, \dots, 20\}$ data points. BA outperforms MA by considerable margins in this regime of little training data.

| | PB/det. | | VI | | MC | | ANP |
|---|---|---|---|---|---|---|---|
| | BA | MA (CNP) | BA | MA (LP-NP) | BA | MA | MA + Attention |
| Quadratic 1D, $L = 64$ | **1.42 ± 0.20** | 0.47 ± 0.25 | **1.48 ± 0.05** | −0.32 ± 0.55 | **1.71 ± 0.23** | 1.27 ± 0.06 | 0.69 ± 0.08 |
| Quadratic 3D, $L = 128$ | **−2.46 ± 0.12** | −2.73 ± 0.10 | **−2.53 ± 0.07** | −3.45 ± 0.12 | **−1.79 ± 0.07** | −2.14 ± 0.05 | −3.08 ± 0.02 |

| BA+PB | MA+det. | BA+VI | MA+VI | BA+MC | MA+MC | ANP |
|---|---|---|---|---|---|---|
| 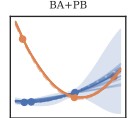 | 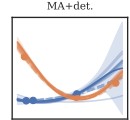 | 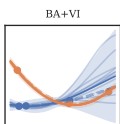 | 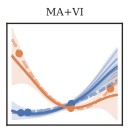 | 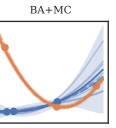 | 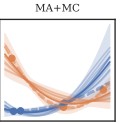 | 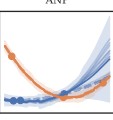 |

Figure 3: Predictions on two instances (dashed lines) of the 1D quadratic function class, given $N = 3$ context data points (circles). We show mean and standard deviation predictions (solid line, shaded area), and 10 function samples (AR samples for deterministic methods). Cf. also App. 7.6.

terestingly, despite employing a factorized Gaussian approximation, our deterministic PB approximation performs at least on-par with the traditional VI approximation which tends to perform particularly poorly for small context sets, reflecting the intricacies discussed in Sec. 4.2. As expected, the MC approximation yields the best results in terms of predictive performance, as it is more expressive than the deterministic approaches and does not share the problems of the VI approach. As shown in Tab. 2 and Tab. 9, App. 7.6, our proposed PB likelihood approximation is much cheaper to evaluate compared to both sampling-based approaches which require multiple forward passes per prediction. We further observe that BA tends to require smaller encoder and decoder networks as it is more efficient at propagating context information to the latent state as discussed in Sec. 4.1. The hybrid ANP approach is competitive only on the Matern-5/2 function class. Yet, we refer the reader to Tab. 10, App. 7.6, demonstrating that the attention mechanism greatly improves performance in terms of MSE.

Table 2: Relative evaluation runtimes and #parameters of the optimized network architectures on RBF GP. Also cf. Tab. 9.

| | PB/det. | | VI | | MC | |
|---|---|---|---|---|---|---|
| | BA | MA (CNP) | BA | MA (LP-NP) | BA | MA |
| Runtime | 1 | 1.4 | 18 | 25 | 32 | 27 |
| #Parameters | 72k | 96k | 63k | 77k | 122k | 153k |

**Quadratic Functions.** We further seek to study the performance of BA with very limited amounts of training data. To this end, we consider two quadratic function classes, each parametrized by three real parameters from which we generate limited numbers $L$ of training tasks. The first function class is defined on a one-dimensional domain, i.e., $x \in \mathbb{R}$, and we choose $L = 64$, while the second function class, as proposed by Perrone et al. (2018), is defined on $x \in \mathbb{R}^3$ with $L = 128$, cf. App. 7.5.1. As shown in Tab. 3, BA again consistently outperforms MA, often by considerably large margins, underlining the efficiency of our Bayesian approach to aggregation in the regime of little training data. On the 1D task, all likelihood approximations perform approximately on-par in combination with BA, while MC outperforms both on the more complex 3D task. Fig. 3 compares prediction qualities.

**Dynamics of a Furuta Pendulum.** We study BA on a realistic dataset given by the simulated dynamics of a rotary inverted pendulum, better known as the Furuta pendulum (Furuta et al., 1992), which is a highly non-linear dynamical system, consisting of an actuated arm rotating in the horizontal plane with an attached pendulum rotating freely in the vertical plane, parametrized by two masses, three lengths, and two damping constants. The regression task is defined as the one-step-ahead prediction of the four-dimensional system state with a step-size of $\Delta t = 0.1\,\mathrm{s}$, as detailed in App. 7.5.1. The results (Tab. 4) show that BA improves predictive performance also on complex, non-synthetic regression tasks with higher-dimensional input- and output spaces. Further, they are consistent 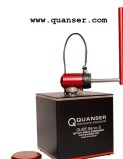 with our previous findings regarding the likelihood approximations, with MC being strongest in terms of predictive likelihood, followed by our efficient deterministic alternative PB.

**2D Image Completion.** We consider a 2D image completion experiment where the inputs $x$ are pixel locations in images showing handwritten digits, and we regress onto the corresponding pixel intensities $y$, cf. App. 7.6. Interestingly, we found that architectures without deterministic paths were not able to solve this task reliably which is why we only report results for deterministic models.

Table 4: Posterior predictive log-likelihood on the dynamics of a Furuta pendulum, averaged over context sets with $N \in \{0, 1, \ldots, 20\}$ state transitions. BA performs favorably on this real-world task.

| | PB/det. | | VI | | MC | | ANP |
|---|---|---|---|---|---|---|---|
| | BA | MA (CNP) | BA | MA (LP-NP) | BA | MA | MA + Attention |
| Furuta Dynamics | $\mathbf{7.50 \pm 0.27}$ | $7.06 \pm 0.12$ | $\mathbf{7.32 \pm 0.18}$ | $5.57 \pm 0.21$ | $\mathbf{8.25 \pm 0.33}$ | $7.55 \pm 0.24$ | $4.74 \pm 0.16$ |

Table 6: Comparison of the posterior predictive log-likelihood of our BA with traditional MA, combined with a self-attention (SA) mechanism in the encoder (BA does not use an SA mechanism), using the PB and MC likelihood approximations. We provide results for Laplace SA (L-SA), dot-product SA (DP-SA), and mulihead SA (MH-SA) and repeat the results for BA and MA without SA ("no SA"). While L-SA and DP-SA do not increase predictive performance compared to MA without SA, MH-SA results in significant improvements. Nevertheless, vanilla BA still performs better or at least on-par, while being computationally more efficient.

| | BA + PB | MA + PB | | | | BA + MC | MA + MC | | | |
|---|---|---|---|---|---|---|---|---|---|---|
| | no SA | no SA | L-SA | DP-SA | MH-SA | no SA | no SA | L-SA | DP-SA | MH-SA |
| RBF GP | $\mathbf{1.37 \pm 0.15}$ | $0.94 \pm 0.04$ | $0.74 \pm 0.06$ | $0.89 \pm 0.04$ | $\mathbf{1.46 \pm 0.14}$ | $\mathbf{1.62 \pm 0.05}$ | $1.07 \pm 0.05$ | $0.93 \pm 0.05$ | $0.98 \pm 0.03$ | $1.44 \pm 0.09$ |
| Weakly Periodic GP | $\mathbf{1.13 \pm 0.08}$ | $0.76 \pm 0.02$ | $0.59 \pm 0.02$ | $0.71 \pm 0.02$ | $\mathbf{1.13 \pm 0.15}$ | $\mathbf{1.30 \pm 0.06}$ | $0.85 \pm 0.04$ | $0.77 \pm 0.03$ | $0.82 \pm 0.03$ | $\mathbf{1.29 \pm 0.04}$ |
| Matern-5/2 GP | $-0.50 \pm 0.07$ | $-0.68 \pm 0.01$ | $-1.03 \pm 0.01$ | $-0.76 \pm 0.01$ | $-0.64 \pm 0.01$ | $\mathbf{-0.33 \pm 0.01}$ | $-0.90 \pm 0.15$ | $-0.80 \pm 0.02$ | $-0.86 \pm 0.01$ | $-0.59 \pm 0.03$ |
| Quadratic 1D, $L = 64$ | $\mathbf{1.42 \pm 0.20}$ | $0.47 \pm 0.25$ | $0.15 \pm 0.32$ | $0.47 \pm 0.24$ | $\mathbf{1.49 \pm 0.11}$ | $\mathbf{1.71 \pm 0.23}$ | $1.27 \pm 0.06$ | $1.19 \pm 0.09$ | $1.32 \pm 0.14$ | $\mathbf{1.66 \pm 0.12}$ |
| Quadratic 3D, $L = 128$ | $-2.46 \pm 0.12$ | $-2.73 \pm 0.10$ | $-2.94 \pm 0.41$ | $-2.95 \pm 0.13$ | $-2.13 \pm 0.25$ | $\mathbf{-1.79 \pm 0.07}$ | $-2.14 \pm 0.05$ | $-2.19 \pm 0.11$ | $-2.18 \pm 0.07$ | $\mathbf{-1.71 \pm 0.05}$ |
| Furuta Dynamics | $\mathbf{7.50 \pm 0.27}$ | $7.06 \pm 0.12$ | $7.13 \pm 0.12$ | $7.04 \pm 0.20$ | $\mathbf{7.40 \pm 0.46}$ | $\mathbf{8.25 \pm 0.33}$ | $7.55 \pm 0.24$ | $7.80 \pm 0.13$ | $7.67 \pm 0.14$ | $\mathbf{8.39 \pm 0.20}$ |

As shown in Tab. 5, BA improves performance in comparison to MA by a large margin. This highlights that BA's ability to quantify the information content of a context tuple is particularly beneficial on this task, as, e.g., pixels in the middle area of the images typically convey more information about the identity of the digit than pixels located near the borders.

Table 5: Predictive log-likelihood on a 2D image completion task on MNIST, averaged over $N \in \{0, 1, \ldots, 392\}$ context pixels.

| | PB/det. | | ANP |
|---|---|---|---|
| | BA | MA (CNP) | MA + Attention |
| 2D Image Completion | $\mathbf{2.75 \pm 0.20}$ | $2.05 \pm 0.36$ | $1.62 \pm 0.03$ |

**Self-attentive Encoders.** Another interesting baseline for BA is MA, combined with a self-attention (SA) mechanism in the encoder. Indeed, similar to BA, SA yields non-uniform weights for the latent observations $r_n$, where a given weight is computed from some form of pairwise spatial relationship with all other latent observations in the context set (cf. App. 7.3 for a detailed discussion). As BA's weight for $r_n$ only depends on $(x_n, y_n)$ itself, BA is computationally more efficient: SA scales like $\mathcal{O}(N^2)$ in the number $N$ of context tuples while BA scales like $\mathcal{O}(N)$, and, furthermore, SA does not allow for efficient incremental updates while this is possible for BA, cf. Eq. (9). Tab. 6 shows a comparison of BA with MA in combination with various different SA mechanisms in the encoder. We emphasize that we compare against BA in its vanilla form, i.e., BA does not use SA in the encoder. The results show that Laplace SA and dot-product SA do not improve predictive performance compared to vanilla MA, while multihead SA yields significantly better results. Nevertheless, vanilla BA still performs better or at least on-par and is computationally more efficient. While being out of the scope of this work, according to these results, a combination of BA with SA seems promising if computational disadvantages can be accepted in favour of increased predictive performance, cf. App. 7.3.

## 6 Conclusion and Outlook

We proposed a novel Bayesian Aggregation (BA) method for NP-based models, combining context aggregation and hidden parameter inference in one holistic mechanism which enables efficient handling of task ambiguity. BA is conceptually simple, compatible with existing NP-based model architectures, and consistently improves performance compared to traditional mean aggregation. It introduces only marginal computational overhead, simplifies the architectures in comparison to existing CLV models (no $\bar{r}$-to-$z$-networks), and tends to require less complex encoder and decoder network architectures. Our experiments further demonstrate that the VI likelihood approximation traditionally used to train NP-based models should be abandoned in favor of a MC-based approach, and that our proposed PB likelihood approximation represents an efficient deterministic alternative with strong predictive performance. We believe that a range of existing models, e.g., the ANP or NPs with self-attentive encoders, can benefit from BA, especially when a reliable quantification of uncertainty is crucial. Also, more complex Bayesian aggregation models are conceivable, opening interesting avenues for future research.

ACKNOWLEDGMENTS

We thank Philipp Becker, Stefan Falkner, and the anonymous reviewers for valuable remarks and discussions which greatly improved this paper.

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

## 7 APPENDIX

We present the derivation of the Bayesian aggregation update equations (Eqs. (8), (9)) in more detail. To foster reproducibility, we describe all experimental settings as well as the hyperparameter optimization procedure used to obtain the results reported in Sec. 5, and publish the source code online.[5] We further provide additional experimental results and visualizations of the predictions of the compared architectures.

### 7.1 DERIVATION OF THE BAYESIAN AGGREGATION UPDATE EQUATIONS

We derive the full Bayesian aggregation update equations without making any factorization assumptions. We start from a Gaussian observation model of the form

$$p\left(r_n \mid z\right) \equiv \mathcal{N}\left(r_n \mid z, \Sigma_{r_n}\right), \quad r_n = \text{enc}_{r,\phi}\left(x_n^c, y_n^c\right), \quad \Sigma_{r_n} = \text{enc}_{\Sigma_r,\phi}\left(x_n^c, y_n^c\right), \quad (12)$$

where $r_n$ and $\Sigma_{r_n}$ are learned by the encoder network. If we impose a Gaussian prior in the latent space, i.e.,

$$p\left(z\right) \equiv \mathcal{N}\left(z \mid \mu_{z,0}, \Sigma_{z,0}\right), \quad (13)$$

we arrive at a Gaussian aggregation model which allows to derive the parameters of the posterior distribution, i.e., of

$$q_\phi\left(z \mid \mathcal{D}^c\right) = \mathcal{N}\left(z \mid \mu_z, \Sigma_z\right) \quad (14)$$

in closed form using standard Gaussian conditioning (Bishop, 2006):

$$\Sigma_z = \left[\left(\Sigma_{z,0}\right)^{-1} + \sum_{n=1}^{N}\left(\Sigma_{r_n}\right)^{-1}\right]^{-1}, \quad (15a)$$

$$\mu_z = \mu_{z,0} + \Sigma_z \sum_{n=1}^{N}\left(\Sigma_{r_n}\right)^{-1}\left(r_n - \mu_{z,0}\right). \quad (15b)$$

As the latent space $z$ is shaped by the encoder network, it will find a space where the following factorization assumptions work well (given $d_z$ is large enough):

$$\Sigma_{r_n} = \text{diag}\left(\sigma_{r_n}^2\right), \quad \sigma_{r_n}^2 = \text{enc}_{\sigma_r^2,\phi}\left(x_n^c, y_n^c\right), \quad \Sigma_{z,0} = \text{diag}\left(\sigma_{z,0}^2\right). \quad (16)$$

This yields a factorized posterior, i.e.,

$$q_\phi\left(z \mid \mathcal{D}^c\right) = \mathcal{N}\left(z \mid \mu_z, \text{diag}\left(\sigma_z^2\right)\right), \quad (17)$$

with

$$\sigma_z^2 = \left[\left(\sigma_{z,0}^2\right)^{\ominus} + \sum_{n=1}^{N}\left(\sigma_{r_n}^2\right)^{\ominus}\right]^{\ominus}, \quad (18a)$$

$$\mu_z = \mu_{z,0} + \sigma_z^2 \odot \sum_{n=1}^{N}\left(r_n - \mu_{z,0}\right) \oslash \left(\sigma_{r_n}^2\right). \quad (18b)$$

Here $\ominus$, $\odot$ and $\oslash$ denote element-wise inversion, product, and division, respectively. This is the result Eq. (8) from the main part of this paper.

### 7.2 DISCUSSION OF VI LIKELIHOOD APPROXIMATION

To highlight the limitations of the VI approximation, we note that decoder networks of models employing the PB or the MC likelihood approximation are provided with the same context information at training and test time: the latent variable (which is passed on to the decoder in the form of latent samples $z$ (for MC) or in the form of parameters $\mu_z$, $\sigma_z^2$ describing the latent distribution (for PB)) is in both cases conditioned only on the context set $\mathcal{D}^c$. In contrast, in the variational approximation Eq. (2), the expectation is w.r.t. $q_\phi$, conditioned on the union of the context set $\mathcal{D}^c$ and the target set $\mathcal{D}^t$. As $\mathcal{D}^t$ is not available at test time, this introduces a mismatch between how the model is trained

---

[5]https://github.com/boschresearch/bayesian-context-aggregation

and how it is used at test time. Indeed, the decoder is trained on samples from $q_\phi\left(z \mid \mathcal{D}^c \cup \mathcal{D}^t\right)$ but evaluated on samples from $q_\phi\left(z \mid \mathcal{D}^c\right)$. This is not a serious problem when the model is evaluated on context sets with sizes large enough to allow accurate approximations of the true latent posterior distribution. Small context sets, however, usually contain too little information to infer $z$ reliably. Consequently, the distributions $q_\phi\left(z \mid \mathcal{D}^c\right)$ and $q_\phi\left(z \mid \mathcal{D}^c \cup \mathcal{D}^t\right)$ typically differ significantly in this regime. Hence, incentivizing the decoder to yield meaningful predictions on small context sets requires intricate and potentially expensive additional sampling procedures to choose suitable target sets $\mathcal{D}^t$ during training. As a corner case, we point out that it is not possible to train the decoder on samples from the latent prior, because the right hand side of Eq. (2) vanishes for $\mathcal{D}^c = \mathcal{D}^t = \varnothing$.

## 7.3 Self-Attentive Encoder Architectures

Kim et al. (2019) propose to use attention-mechanisms to improve the quality of NP-based regression. In general, given a set of key-value pairs $\{(x_n, y_n)\}_{n=1}^N$, $x_n \in \mathbb{R}^{d_x}$, $y_n \in \mathbb{R}^{d_y}$, and a query $x^* \in \mathbb{R}^{d_x}$, an attention mechanism $\mathcal{A}$ produces a weighted sum of the values, with the weights being computed from the keys and the query:

$$\mathcal{A}\left(\{(x_n, y_n)\}_{n=1}^N, x^*\right) = \sum_{n=1}^N w\left(x_n, x^*\right) y_n. \tag{19}$$

There are several types of attention mechanisms proposed in the literature (Vaswani et al., 2017), each defining a specific form of the weights. *Laplace attention* adjusts the weights according to the spatial distance of keys and query:

$$w_{\mathrm{L}}\left(x_n, x^*\right) \propto \exp\left(-||x_n - x^*||_1\right). \tag{20}$$

Similarly, *dot-product attention* computes

$$w_{\mathrm{DP}}\left(x_n, x^*\right) \propto \exp\left(x_n^T x^* / \sqrt{d_x}\right). \tag{21}$$

A more complex mechanism is *multihead attention*, which employs a set of $3H$ learned linear mappings $\left\{\mathcal{L}_h^K\right\}_{h=1}^H$, $\left\{\mathcal{L}_h^V\right\}_{h=1}^H$, $\left\{\mathcal{L}_h^Q\right\}_{h=1}^H$, where $H$ is a hyperparameter. For each $h$, these mappings are applied to keys, values, and queries, respectively. Subsequently, dot-product attention is applied to the set of transformed key-value pairs and the transformed query. The resulting $H$ values are then again combined by a further learned linear mapping $\mathcal{L}^O$ to obtain the final result.

*Self-attention* (SA) is defined by setting the set of queries equal to the set of keys. Therefore, SA produces again a set of $N$ weighted values. Combining SA with an NP-encoder, i.e., applying SA to the set $\{f_x(x_n), r_n\}_{n=1}^N$ of inputs $x_n$ and corresponding latent observations $r_n$ (where we also consider a possible nonlinear transformation $f_x$ of the inputs) and subsequently applying MA yields an interesting baseline for our proposed BA. Indeed, similar to BA, SA computes a weighted sum of the latent observations $r_n$. Note, however, that SA weighs each latent observation according to some form of spatial relationship of the corresponding input with all other latent observations in the context set. In contrast, BA's weight for a given latent observation is based only on features computed from the context tuple corresponding to this very latent observation and allows to incorporate an estimation of the amount of information contained in the context tuple into the aggregation (cf. Sec. 4.1). This leads to several computational advantages of BA over SA: (i) SA scales quadratically in the number $N$ of context tuples, as it has to be evaluated on all $N^2$ pairs of context tuples. In contrast, BA scales linearly with $N$. (ii) BA allows for efficient incremental updates when context data arrives sequentially (cf. Eq. (9)), while using SA does not provide this possibility: it requires to store and encode the whole context set $\mathcal{D}^c$ at once and to subsequently aggregate the whole set of resulting (SA-weighted) latent observations.

The results in Tab. 6, Sec. 5 show that multihead SA leads to significant improvements in predictive performance compared to vanilla MA. Therefore, a combination of BA with self-attentive encoders seems promising in situations where computational disadvantages can be accepted in favour of increased predictive performance. Note that BA relies on a second encoder output $\sigma_{r_n}^2$ (in addition to the latent observation $r_n$) which assesses the information content in each context tuple $(x_n, y_n)$. As each SA-weighted $r_n$ is informed by the other latent observations in the context set, obviously, one would have to also process the set of $\sigma_{r_n}^2$ in a manner consistent with the SA-weighting. We leave such a combination of SA and BA for future research.

Table 7: Comparison of the predictive log-likelihood of NP-based architectures with two simple GP-based baselines, (i) Vanilla GP (optimizes the hyperparameters individually on each target task and ignores the source data) (ii) Multi-task GP (optimizes one set of hyperparameters on all source tasks and uses them without further adaptation on the target tasks). Both GP implementations use RBF-kernels. As in the main text, we average performance over context sets with sizes $N \in \{0, ..., 64\}$ for RBF GP and $N \in \{0, ..., 20\}$ for the other experiments. Multi-task GP constitutes the optimal model (assuming it fits the hyperparameters perfectly) for the RBF GP experiment, which explains its superior performance. On the Quadratic 1D experiment, Multi-task GP still performs better than the other methods as this function class shows a relatively low degree of variability. In contrast, on more complex experiments like Quadratic 3D and the Furuta dynamics, none of the GP variants is able to produce meaningful results given the small budget of at most 20 context points, while NP-based methods produce predictions of high quality as they incorporate the source data more efficiently.

| | NPs with MC-loss | | GP | |
| | BA | MA | Vanilla | Multi-task |
|---|---|---|---|---|
| RBF GP | $1.62 \pm 0.05$ | $1.07 \pm 0.05$ | $1.96$ | $\mathbf{2.99}$ |
| Quadratic 1D, $L = 64$ | $1.71 \pm 0.23$ | $1.27 \pm 0.06$ | $-1.56$ | $\mathbf{2.11}$ |
| Quadratic 3D, $L = 128$ | $\mathbf{-1.79 \pm 0.07}$ | $-2.14 \pm 0.05$ | $-472.76$ | $-173.78$ |
| Furuta Dynamics | $\mathbf{8.25 \pm 0.33}$ | $7.55 \pm 0.24$ | $-6.16$ | $-2.47$ |

## 7.4 NEURAL PROCESS-BASED MODELS IN THE CONTEXT OF SCALABLE PROBABILISTIC REGRESSION

We discuss in more detail how NP-based models relate to other existing methods for (scalable) probabilistic regression, such as (multi-task) GPs (Rasmussen and Williams, 2005; Bardenet et al., 2013; Yogatama and Mann, 2014; Golovin et al., 2017), Bayesian neural networks (BNNs) (MacKay, 1992; Gal and Ghahramani, 2016), and DeepGPs (Damianou and Lawrence, 2013).

NPs are motivated in Garnelo et al. (2018a;b), Kim et al. (2019), as well as in our Sec. 1, as models which combine the computational efficiency of neural networks with well-calibrated uncertainty estimates (like those of GPs). Indeed, NPs scale linearly in the number $N$ of context and $M$ of target data points, i.e., like $\mathcal{O}(N + M)$, while GPs scale like $\mathcal{O}(N^3 + M^2)$. Furthermore, NPs are shown to exhibit well-calibrated uncertainty estimates. In this sense, NPs can be counted as members of the family of scalable probabilistic regression methods.

A central aspect of NP training which distinguishes NPs from a range of standard methods is that they are trained in a multi-task fashion (cf. Sec. 3). This means that NPs rely on data from a set of related source tasks from which they automatically learn powerful priors and the ability to adapt quickly to unseen target tasks. This multi-task training procedure of NPs scales linearly in the number $L$ of source tasks, which makes it possible to train these architectures on large amounts of source data. Applying GPs in such a multi-task setting can be challenging, especially for large numbers of source tasks. Similarly, BNNs as well as DeepGPs are in their vanilla forms specifically designed for the single-task setting. Therefore, GPs, BNNs, and DeepGPs are not directly applicable in the NP multi-task setting, which is why they are typically not considered as baselines for NP-based models, as discussed in (Kim et al., 2019).

The experiments presented in Garnelo et al. (2018a;b) and Kim et al. (2019) focus mainly on evaluating NPs in the context of few-shot probabilistic regression, i.e., on demonstrating the data-efficiency of NPs on the target task after training on data from a range of source tasks. In contrast, the application of NPs in situations with large ($> 1000$) numbers of context/target points per task has to the best of our knowledge not yet been investigated in detail in the literature. Furthermore, it has not been studied how to apply NPs in situations where only a single or very few source tasks are available. The focus of our paper is a clear-cut comparison of the performance of our BA with traditional MA in the context of NP-based models. Therefore, we also consider experiments similar to those presented in (Garnelo et al., 2018a;b; Kim et al., 2019) and leave further comparisons with existing methods for (multi-task) probabilistic regressions for future work.

Nevertheless, to illustrate this discussion, we provide two simple GP-based baseline methods: (i) a vanilla GP, which optimizes the hyperparameters on each target task individually and does not use

the source data, and (ii) a naive but easily interpretable example of a multi-task GP, which optimizes one set of hyperparameters on all source tasks and uses it for predictions on the target tasks without further adaptation. The results in Tab. 7 show that those GP-based models can only compete with NPs on function classes where either the inductive bias as given by the kernel functions fits the data well (RBF GP), or on function classes which exhibit a relatively low degree of variablity (Quadratic 1D). On more complex function classes, NPs produce predictions of much better quality, as they incorporate the source data more efficiently.

### 7.5 EXPERIMENTAL DETAILS

We provide details about the data sets as well as about the experimental setup used in our experiments in Sec. 5.

#### 7.5.1 DATA GENERATION

In our experiments, we use several classes of functions to evaluate the architectures under consideration. To generate training data from these function classes, we sample $L$ random tasks (as described in Sec. 5), and $N_{\text{tot}}$ random input locations $x$ for each task. For each minibatch of training tasks, we uniformly sample a context set size $N \in \{n_{\min}, \ldots, n_{\max}\}$ and use a random subset of $N$ data points from each task as context sets $\mathcal{D}^c$. The remaining $M = N_{\text{tot}} - N$ data points are used as the target sets $\mathcal{D}^t$ (cf. App. 7.5.3 for the special case of the VI likelihood approximation). Tab. 8 provides details about the data generation process.

**GP Samples.** We sample one-dimensional functions $f : \mathbb{R} \to \mathbb{R}$ from GP priors with three different stationary kernel functions as proposed by Gordon et al. (2020).

A radial basis functions (RBF) kernel with lenghtscale $l = 1.0$:

$$k_{\text{RBF}}(r) \equiv \exp\left(-0.5r^2\right). \tag{22}$$

A weakly periodic kernel:

$$k_{\text{WP}}(r) \equiv \exp\left(-2\sin(0.5r)^2 - 0.125r^2\right). \tag{23}$$

A Matern-5/2 kernel with lengthscale $l = 0.25$:

$$k_{\text{M5/2}}(r) \equiv \left(1 + \frac{\sqrt{5}r}{0.25} + \frac{5r^2}{3 \cdot 0.25^2}\right) \exp\left(-\frac{\sqrt{5}r}{0.25}\right). \tag{24}$$

**Quadratic Functions.** We consider two classes of quadratic functions. The first class $f^{Q,1D} : \mathbb{R} \to \mathbb{R}$ is defined on a one-dimensional domain and parametrized by three parameters $a, b, c \in \mathbb{R}$:

$$f^{Q,1D}(x) \equiv a^2(x + b)^2 + c. \tag{25}$$

The second class $f^{Q,3D} : \mathbb{R}^3 \to \mathbb{R}$ is defined on a three-dimensional domain and also parametrized by three parameters $a, b, c \in \mathbb{R}$:

$$f^{Q,3D}(x_1, x_2, x_3) \equiv 0.5a\left(x_1^2 + x_2^2 + x_3^2\right) + b(x_1 + x_2 + x_3) + 3c. \tag{26}$$

This function class was proposed in Perrone et al. (2018).

For both function classes we add Gaussian noise with standard deviation $\sigma_n$ to the evaluations, cf. Tab. 8.

**Furuta Pendulum Dynamics.** We consider a function class obtained by integrating the non-linear equations of motion governing the dynamics of a Furuta pendulum (Furuta et al., 1992; Cazzolato and Prime, 2011) for a time span of $\Delta t = 0.1\,\text{s}$. More concretely, we consider the mapping

$$\Theta(t) \to \Theta(t + \Delta t) - \Theta(t), \tag{27}$$

Table 8: Input spaces and parameters used to generate data for training and testing the architectures discussed in the main part of this paper. $U(a, b)$ denotes the uniform distribution on the interval $[a, b]$, and, likewise $U\{a, a+n\}$ denotes the uniform distribution on the set $\{a, a+1, \ldots, a+n\}$.

| Symbol | Description | Value/Sampling distribution |
|---|---|---|
| **GP Samples** | | |
| $x$ | Input | $U(-2.0, +2.0)$ |
| $N_{\text{tot}}$ | Number of data points per task | 128 |
| $\{n_{\min}, \ldots n_{\max}\}$ | Context set sizes during training | $\{3, \ldots, 50\}$ |
| **1D Quadratic Functions** | | |
| $x$ | Input | $U(-1.0, +1.0)$ |
| $a$ | Parameter | $U(-0.5, +1.5)$ |
| $b$ | Parameter | $U(-0.9, +0.9)$ |
| $c$ | Parameter | $U(-1.0, +1.0)$ |
| $\sigma_n$ | Noise standard deviation | 0.01 |
| $N_{\text{tot}}$ | Number of data points per task | 128 |
| $\{n_{\min}, \ldots n_{\max}\}$ | Context set sizes during training | $U\{0, \ldots, 20\}$ |
| **3D Quadratic Functions** | | |
| $x_1, x_2, x_3$ | Inputs | $U(-1.0, +1.0)$ |
| $a, b, c$ | Parameters | $U(+0.1, +10.0)$ |
| $\sigma_n$ | Noise standard deviation | 0.01 |
| $N_{\text{tot}}$ | Number of data points per task | 128 |
| $\{n_{\min}, \ldots n_{\max}\}$ | Context set sizes during training | $U\{0, \ldots, 20\}$ |
| **Furuta Dynamics** | | |
| $\theta_{\text{arm}}, \theta_{\text{pend}}$ | Input angles | $U(0.0, 2\pi\,\text{rad})$ |
| $\dot{\theta}_{\text{arm}}, \dot{\theta}_{\text{pend}}$ | Input angular velocities | $U(-2\pi\,\text{rad}/0.5\,\text{s}, 2\pi\,\text{rad}/0.5\,\text{s})$ |
| $m_{\text{arm}}$ | Mass arm | $U(6.0 \cdot 10^{-2}\,\text{kg}, 6.0 \cdot 10^{-1}\,\text{kg})$ |
| $m_{\text{pend}}$ | Mass pendulum | $U(1.5 \cdot 10^{-2}\,\text{kg}, 1.5 \cdot 10^{-1}\,\text{kg})$ |
| $l_{\text{arm}}$ | Length arm | $U(5.6 \cdot 10^{-2}\,\text{m}, 5.6 \cdot 10^{-1}\,\text{m})$ |
| $L_{\text{arm}}$ | Distance joint arm — mass arm | $U(1.0 \cdot 10^{-1}\,\text{m}, 3.0 \cdot 10^{-1}\,\text{m})$ |
| $L_{\text{pend}}$ | Distance joint pend. — mass pend. | $U(1.0 \cdot 10^{-1}\,\text{m}, 3.0 \cdot 10^{-1}\,\text{m})$ |
| $b_{\text{arm}}$ | Damping constant arm | $U(2.0 \cdot 10^{-5}\,\text{Nms}, 2.0 \cdot 10^{-3}\,\text{Nms})$ |
| $b_{\text{pend}}$ | Damping constant pendulum | $U(5.6 \cdot 10^{-5}\,\text{Nms}, 5.6 \cdot 10^{-3}\,\text{Nms})$ |
| $\sigma_{\tau, \text{arm}}$ | Action noise standard dev. arm | $0.5\,\text{Nm}$ |
| $\sigma_{\tau, \text{pend}}$ | Action noise standard dev. pend. | $0.5\,\text{Nm}$ |
| $N_{\text{tot}}$ | Number of data points per task | 256 |
| $\{n_{\min}, \ldots n_{\max}\}$ | Context set sizes during training | $U\{0, \ldots, 20\}$ |
| **2D Image Completion MNIST** | | |
| $x_1, x_2$ | Input pixel locations | $U\{0, 27\}$ (scaled to $[0, 1]$) |
| $N_{\text{tot}}$ | Number of data points per task | $28 \cdot 28$ |
| $\{n_{\min}, \ldots n_{\max}\}$ | Context set sizes during training | $U\{0, \ldots, 28 \cdot 28/2\}$ |

where $\mathbf{\Theta} = \left[\theta_{\text{arm}}(t), \theta_{\text{pend}}(t), \dot{\theta}_{\text{arm}}(t), \dot{\theta}_{\text{pend}}(t)\right]^T$ denotes the four-dimensional vector describing the dynamical state of the Furuta pendulum. The Furuta pendulum is parametrized by seven parameters (two masses, three lengths, two damping constants) as detailed in Tab. 8. During training, we provide $L = 64$ tasks, corresponding to 64 different parameter configurations. We consider the free system and generate noise by applying random torques at each integration time step ($\Delta t_{\text{Euler}} = 0.001\,\text{s}$) to the joints of the arm and pendulum drawn from Gaussian distributions with standard deviations $\sigma_{\tau, \text{pend}}, \sigma_{\tau, \text{arm}}$, respectively.

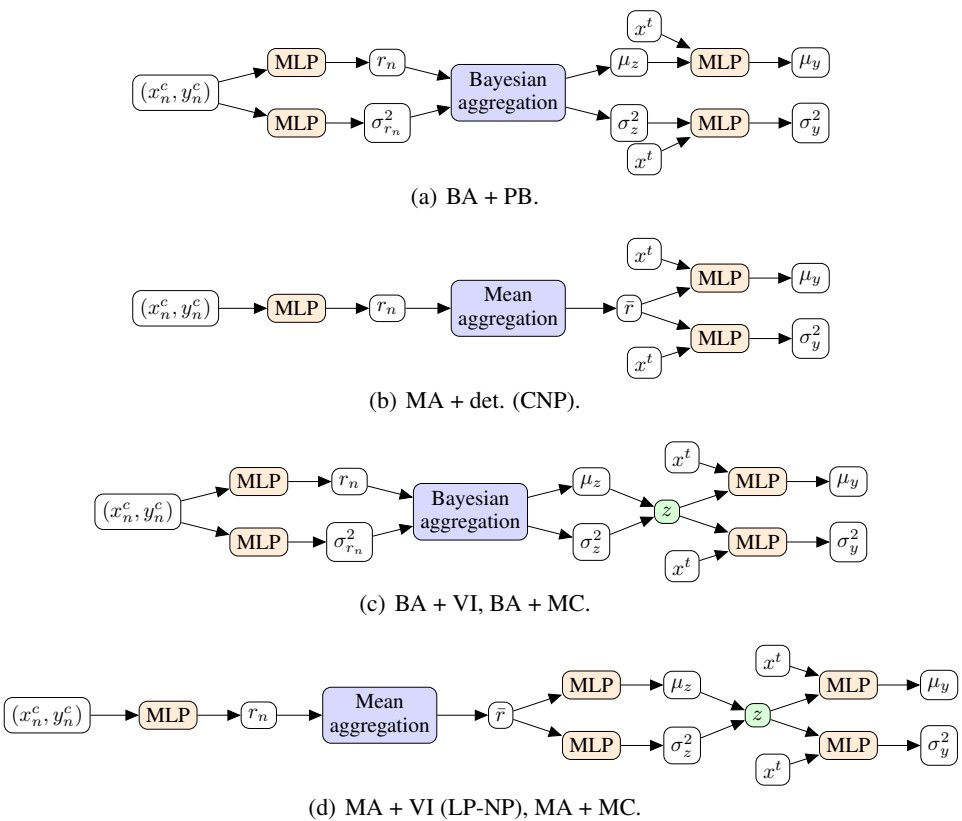

Figure 4: Model architectures used for our experiments in Sec. 5. For the ANP architecture we refer the reader to Kim et al. (2019). Orange rectangles denote MLPs. Blue rectangles denote aggregation operations. Variables in green rectangles are sampled from normal distributions with parameters given by the incoming nodes. To arrive at a fair comparison, we optimize all MLP architectures, the latent space dimensionality $d_z$, as well as the Adam learning rate, individually for all model architectures and all experiments, cf. App. 7.5.3.

**2D Image Completion.** For this task, we use the MNIST database of $28 \times 28$ images of handwritten digits (LeCun and Cortes, 2010), and define 2D functions mapping pixel locations $x_1, x_2 \in \{0, \ldots 27\}$ (scaled to the unit square) to the corresponding pixel intensities $y \in \{0, \ldots, 255\}$ (scaled to the unit interval), cf. Tab. 8. One training task corresponds to one image drawn randomly from the training set (consisting of 60000 images) and for evaluation we use a subset of the test set (consisting of 10000 images).

### 7.5.2 MODEL ARCHITECTURES

We provide the detailed architectures used for the experiments in Sec. 5 in Fig. 4. For ANP we use multihead cross attention and refer the reader to Kim et al. (2019) for details about the architecture.

### 7.5.3 HYPERPARAMETERS AND HYPERPARAMETER OPTIMIZATION

To arrive at a fair comparison of our BA with MA, it is imperative to use optimal model architectures for each aggregation method and likelihood approximation under consideration. Therefore, we optimize the number of hidden layers and the number of hidden units per layer of each encoder and decoder MLP (as shown in Fig. 4), individually for each model architecture and each experiment. For the ANP, we also optimize the multihead attention MLPs. We further optimize the latent space dimensionality $d_z$ and the learning rate of the Adam optimizer. For this hyperparameter optimization, we use the Optuna framework (Akiba et al., 2019) with TPE Sampler and Hyperband pruner (Li et al., 2017). We consistently use a minibatch size of 16. Further, we use $S = 10$ latent samples to evaluate

the MC likelihood approximation during training. To evaluate the VI likelihood approximation, we sample target set sizes between $N_{\text{tot}}$ and $N$ in each training epoch, cf. Tab. 8.

### 7.5.4 Evaluation Procedure

To evaluate the performance of the various model architectures we generate $L = 256$ unseen test tasks with target sets $\mathcal{D}_\ell^t$ consisting of $M = 256$ data points each and compute the average posterior predictive log-likelihood $\frac{1}{L}\frac{1}{M}\sum_{\ell=1}^{L}\log p\left(y_{\ell,1:M}^t \middle| x_{\ell,1:M}^t, \mathcal{D}_\ell^c, \theta\right)$, given context sets $\mathcal{D}_\ell^c$ of size $N$.

Depending on the architecture, we approximate the posterior predictive log-likelihood according to:

- For BA + PB likelihood approximation:

$$\frac{1}{L}\frac{1}{M}\sum_{\ell=1}^{L}\sum_{m=1}^{M}\log p\left(y_{\ell,m}^t \middle| x_{\ell,m}^t, \mu_{z,\ell}, \sigma_{z,\ell}^2, \theta\right). \tag{28}$$

- For MA + deterministic loss (= CNP):

$$\frac{1}{L}\frac{1}{M}\sum_{\ell=1}^{L}\sum_{m=1}^{M}\log p\left(y_{\ell,m}^t \middle| x_{\ell,m}^t, \bar{r}_\ell, \theta\right). \tag{29}$$

- For architectures employing sampling-based likelihood approximations (VI, MC-LL) we report the joint log-likelihood over all data points in a test set, i.e.

$$\frac{1}{L}\frac{1}{M}\sum_{\ell=1}^{L}\log\int q_\phi\left(z_\ell \middle| \mathcal{D}_\ell^c\right)\prod_{m=1}^{M}p\left(y_{\ell,m}^t \middle| x_{\ell,m}^t, z_\ell, \theta\right)\mathrm{d}z_\ell \tag{30}$$

$$\approx \frac{1}{L}\frac{1}{M}\sum_{\ell=1}^{L}\log\frac{1}{S}\sum_{s=1}^{S}\prod_{m=1}^{M}p\left(y_{\ell,m}^t \middle| x_{\ell,m}^t, z_{\ell,s}, \theta\right) \tag{31}$$

$$= -\frac{1}{M}\log S + \frac{1}{L}\frac{1}{M}\sum_{l=1}^{L}\operatorname*{logsumexp}_{s=1}^{S}\left(\sum_{m=1}^{M}\log p\left(y_{\ell,m}^t \middle| x_{\ell,m}^t, z_{\ell,s}, \theta\right)\right), \tag{32}$$

where $z_{\ell,s} \sim q_\phi\left(z \middle| \mathcal{D}_\ell\right)$. We employ $S = 25$ latent samples.

To compute the log-likelihood values given in tables, we additionally average over various context set sizes $N$ as detailed in the main part of this paper.

We report the mean posterior predictive log-likelihood computed in this way w.r.t. 10 training runs with different random seeds together with $95\%$ confidence intervals

Table 9: Relative evaluation runtimes and numbers of parameters of the optimized network architectures on the GP tasks. The deterministic methods (PB, det.) are much more efficient regarding evaluation runtime, as they require only on forward pass per prediction, while the sampling-based approaches (VI, MC) require multiple forward passes (each corresponding to one latent sample) to compute their predictions. We use $S = 25$ latent samples, as described in App. 7.5.4. Furthermore, BA tends to require less complex encoder and decoder network architectures compared to MA, because it represents a more efficient mechanism to propagate information from the context set to the latent state.

| | | PB/det. | | VI | | MC | |
|---|---|---|---|---|---|---|---|
| | | BA | MA (CNP) | BA | MA (LP-NP) | BA | MA |
| RBF GP | Runtime | 1 | 1.4 | 18 | 25 | 32 | 27 |
| | #Parameters | 72k | 96k | 63k | 77k | 122k | 153k |
| Weakly Periodic GP | Runtime | 1 | 1.4 | 11 | 10 | 22 | 15 |
| | #Parameters | 51k | 87k | 48k | 72k | 87k | 89k |
| Matern-5/2 GP | Runtime | 1 | 1.1 | 6.5 | 11 | 15 | 19 |
| | #Parameters | 53k | 100k | 32k | 35k | 108k | 104k |

Table 10: Posterior predictive mean squared error (MSE) on all experiments presented in this paper. We average over the same context set sizes as used to compute the posterior predictive log-likelihood, cf. Sec. 5, and again use $S = 25$ latent samples to compute the mean prediction of sampling-based methods. Our BA consistently improves predictive performance compared to MA not only in terms of likelihood (as shown in Sec. 5), but also in terms of MSE. Furthermore, while ANP tends to perform poorly in terms of likelihood (cf. Sec. 5), it's MSE is improved greatly by the attention mechanism.

| | PB/det. | | VI | | MC | | ANP |
|---|---|---|---|---|---|---|---|
| | BA | MA (CNP) | BA | MA (LP-NP) | BA | MA | MA + Attention |
| RBF GP | $0.0623 \pm 0.0009$ | $0.0687 \pm 0.0010$ | $0.0736 \pm 0.0005$ | $0.0938 \pm 0.0036$ | $0.0637 \pm 0.0007$ | $0.0741 \pm 0.0012$ | $0.0550 \pm 0.0009$ |
| Weakly Periodic GP | $0.0679 \pm 0.0007$ | $0.0761 \pm 0.0014$ | $0.0879 \pm 0.0017$ | $0.1326 \pm 0.0518$ | $0.0677 \pm 0.0008$ | $0.0832 \pm 0.0009$ | $0.0592 \pm 0.0009$ |
| Matern-5/2 GP | $0.2452 \pm 0.0088$ | $0.3021 \pm 0.0035$ | $0.3702 \pm 0.0100$ | $0.6292 \pm 0.1077$ | $0.2321 \pm 0.0019$ | $0.5166 \pm 0.1438$ | $0.1890 \pm 0.0012$ |
| Quadratics 1D, $L = 64$ | $0.1447 \pm 0.0095$ | $0.1513 \pm 0.0091$ | $0.1757 \pm 0.0128$ | $0.1833 \pm 0.0154$ | $0.1473 \pm 0.0107$ | $0.1636 \pm 0.0082$ | $0.1330 \pm 0.0037$ |
| Quadratics 3D, $L = 128$ | $190.5 \pm 1.4$ | $195.4 \pm 1.5$ | $253.1 \pm 18.0$ | $278.1 \pm 40.5$ | $196.8 \pm 2.6$ | $206.7 \pm 5.3$ | $192.5 \pm 2.7$ |
| Furuta Dynamics | $0.1742 \pm 0.0092$ | $0.1989 \pm 0.0095$ | $0.2269 \pm 0.0088$ | $0.2606 \pm 0.0165$ | $0.1758 \pm 0.0124$ | $0.1977 \pm 0.0154$ | $0.1516 \pm 0.0073$ |
| 2D Image Completion | $0.0348 \pm 0.0010$ | $0.0417 \pm 0.0026$ | – | – | – | – | $0.0215 \pm 0.0003$ |

## 7.6 ADDITIONAL EXPERIMENTAL RESULTS

We provide additional experimental results accompanying the experiments presented in Sec. 5:

- Results for relative evaluation runtimes and numbers of parameters of the optimized network architectures on the full GP suite of experiments, cf. Tab. 9.

- The posterior predictive mean squared error on all experiments, cf. Tab. 10.

- The context-size dependent results for the predictive posterior log-likelihood for the 1D and 3D Quadratic experiments, the Furuta dynamics experiment, as well as the 2D image completion experiment, cf. Fig. 5.

- More detailed plots of the predictions on one-dimensional experiments (1D Quadratics (Figs. 6, 7), RBF-GP, (Figs. 8, 9), Weakly Periodic GP (Figs. 10, 11), and Matern-5/2 GP (Figs. 12, 13)).

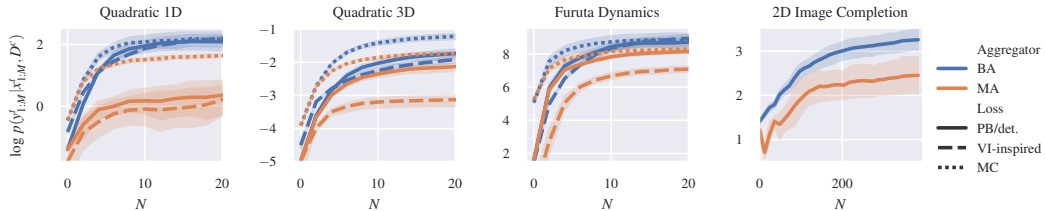

Figure 5: Posterior predictive log-likelihood in dependence of the context set size $N$ for the 1D and 3D Quadratic experiments, the Furuta dynamics experiment as well as the 2D image completion experiment.

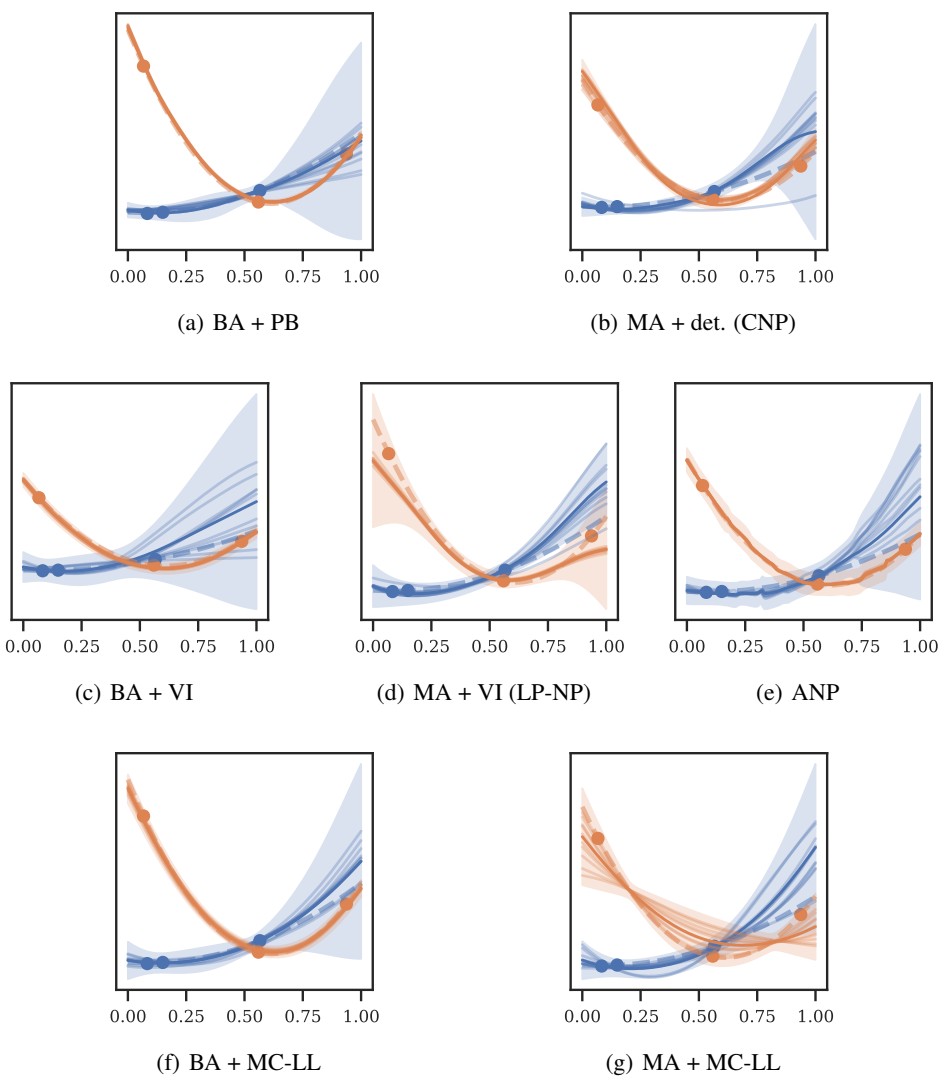

Figure 6: Predictions on two instances (dashed lines) of the 1D quadratic function class, given $N = 3$ context data points (circles). We plot mean and standard deviation (solid line, shaded area) predictions together with 10 function samples (for deterministic methods we employ AR sampling).

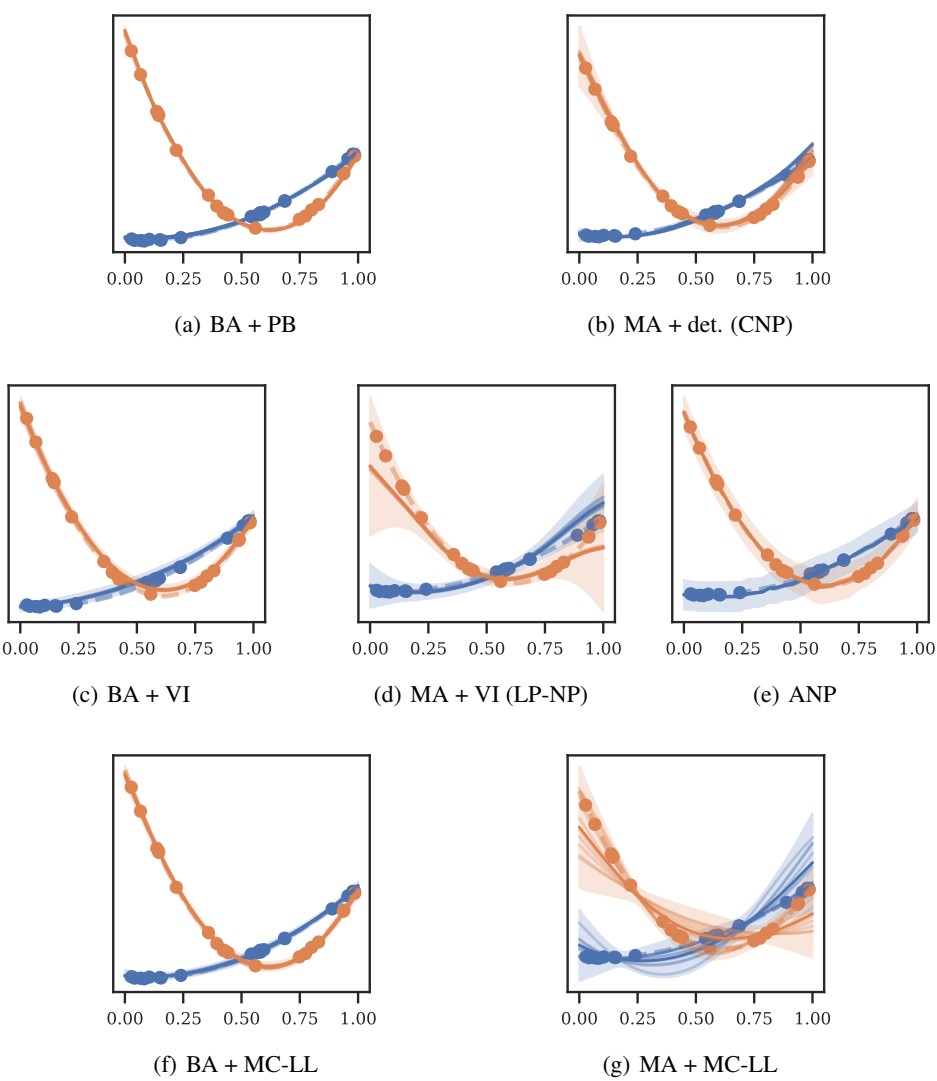

Figure 7: Predictions on two instances (dashed lines) of the 1D quadratic function class, given $N = 19$ context data points (circles). We plot mean and standard deviation (solid line, shaded area) predictions together with 10 function samples (for deterministic methods we employ AR sampling).

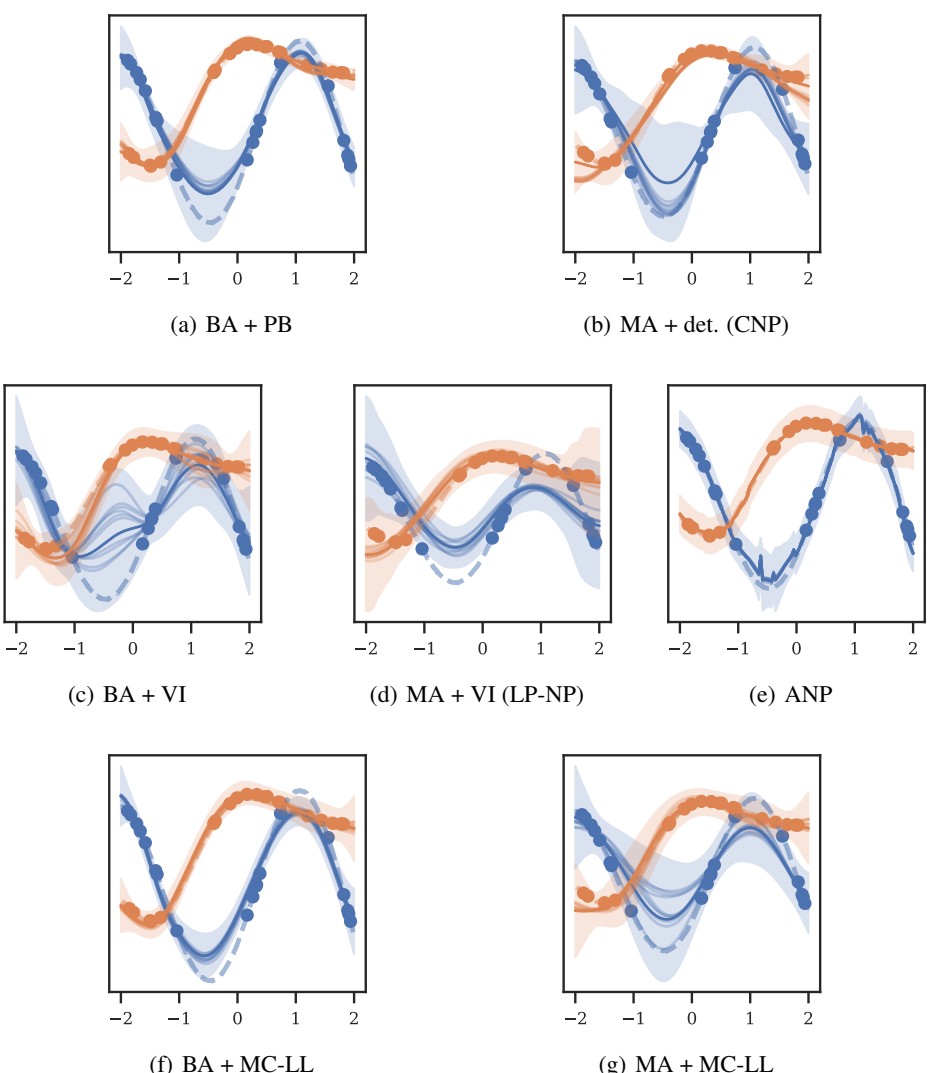

Figure 8: Predictions on two instances (dashed lines) of the RBF GP function class, given $N = 20$ context data points (circles). We plot mean and standard deviation (solid line, shaded area) predictions together with 10 function samples (for deterministic methods we employ AR sampling).

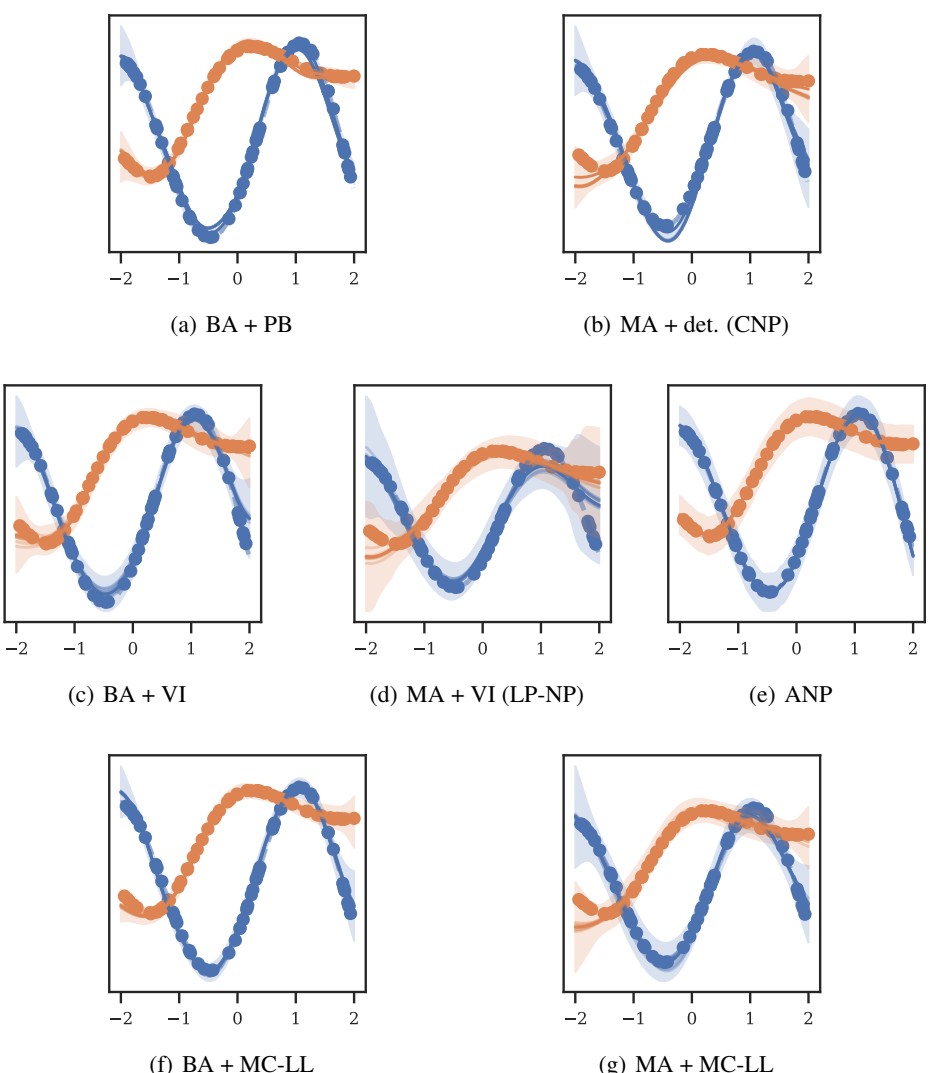

Figure 9: Predictions on two instances (dashed lines) of the RBF GP function class, given $N = 60$ context data points (circles). We plot mean and standard deviation (solid line, shaded area) predictions together with 10 function samples (for deterministic methods we employ AR sampling).

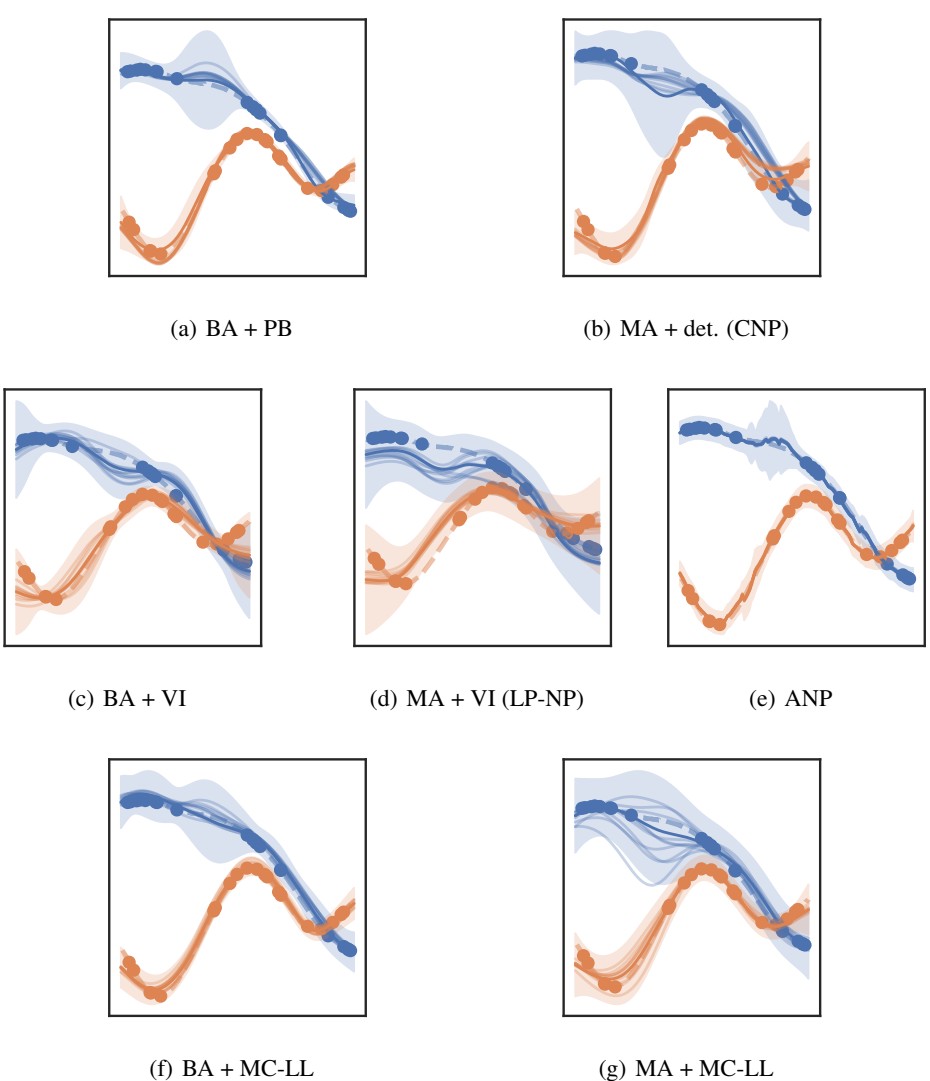

Figure 10: Predictions on two instances (dashed lines) of the Weakly Periodic GP function class, given $N = 20$ context data points (circles). We plot mean and standard deviation (solid line, shaded area) predictions together with 10 function samples (for deterministic methods we employ AR sampling).

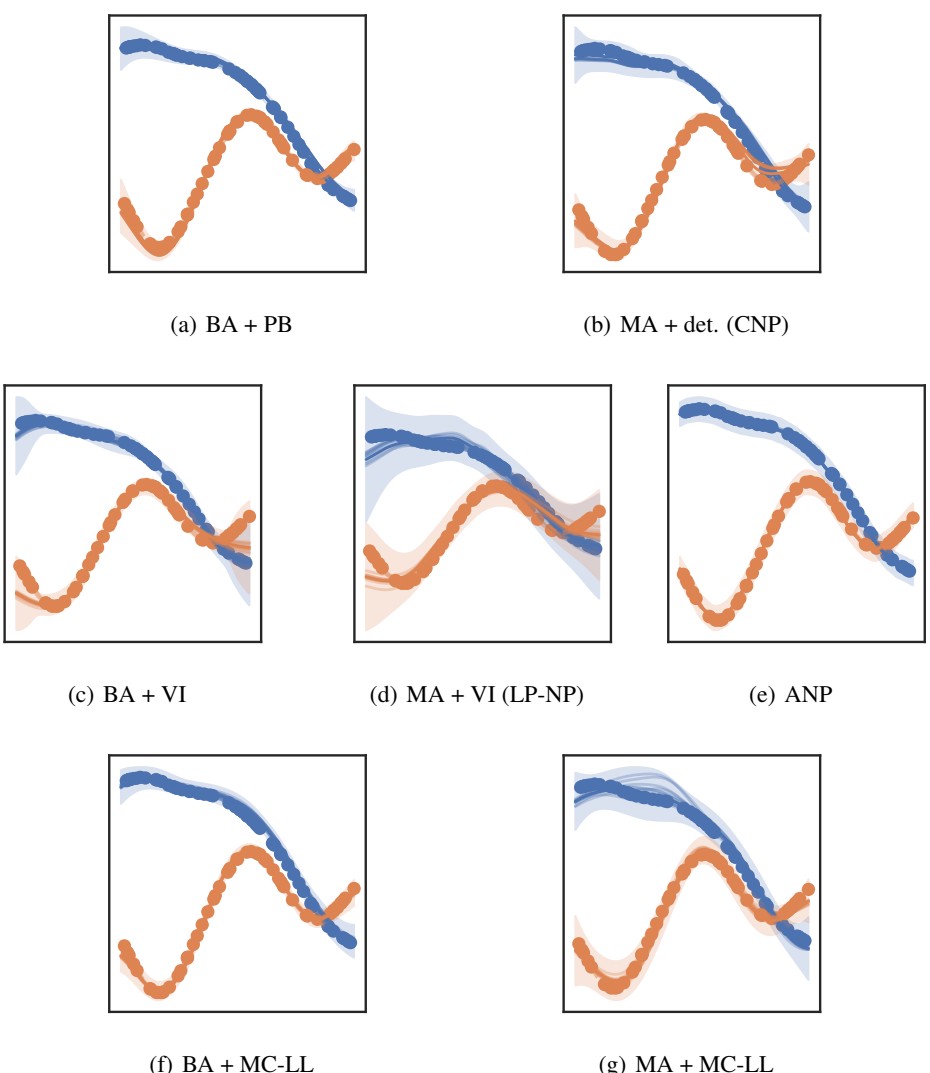

Figure 11: Predictions on two instances (dashed lines) of the Weakly Periodic GP function class, given $N = 60$ context data points (circles). We plot mean and standard deviation (solid line, shaded area) predictions together with 10 function samples (for deterministic methods we employ AR sampling).

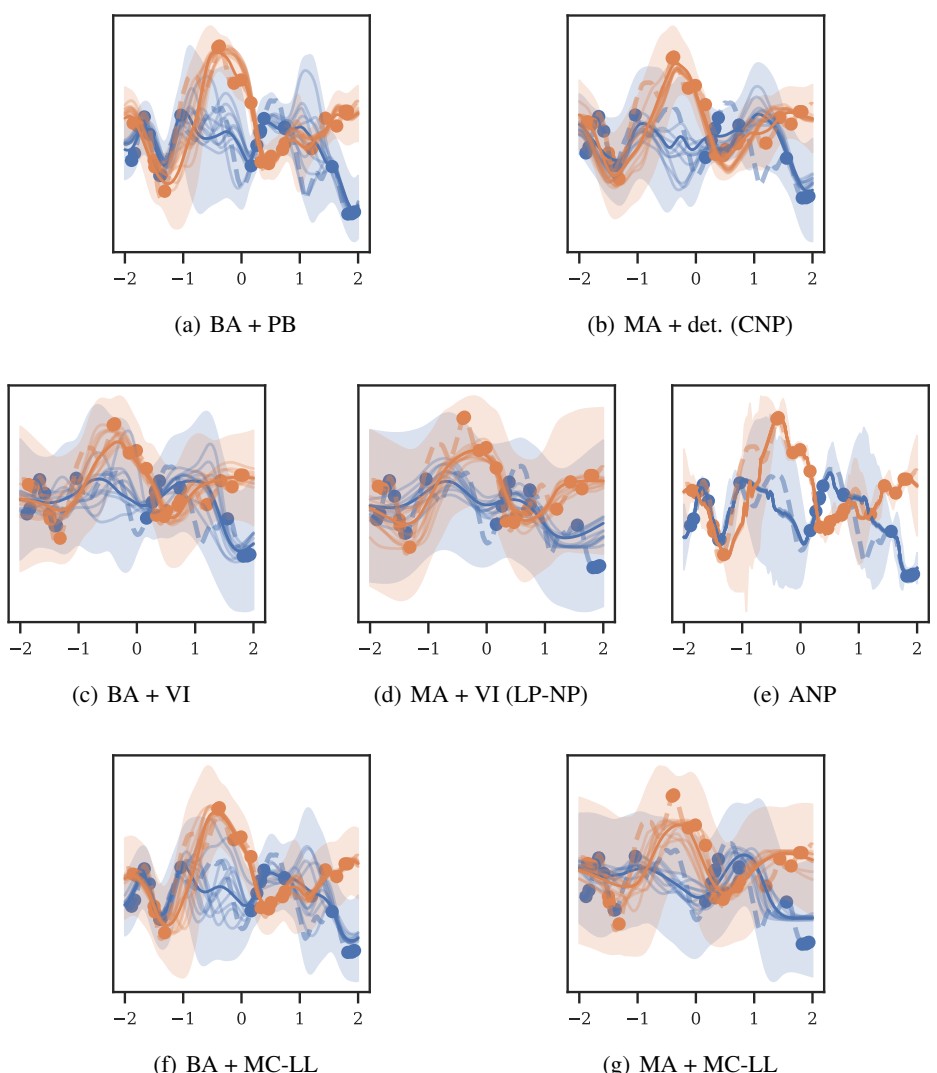

(a) BA + PB  (b) MA + det. (CNP)

(c) BA + VI  (d) MA + VI (LP-NP)  (e) ANP

(f) BA + MC-LL  (g) MA + MC-LL

Figure 12: Predictions on two instances (dashed lines) of the Matern-5/2 GP function class, given $N = 20$ context data points (circles). We plot mean and standard deviation (solid line, shaded area) predictions together with 10 function samples (for deterministic methods we employ AR sampling).

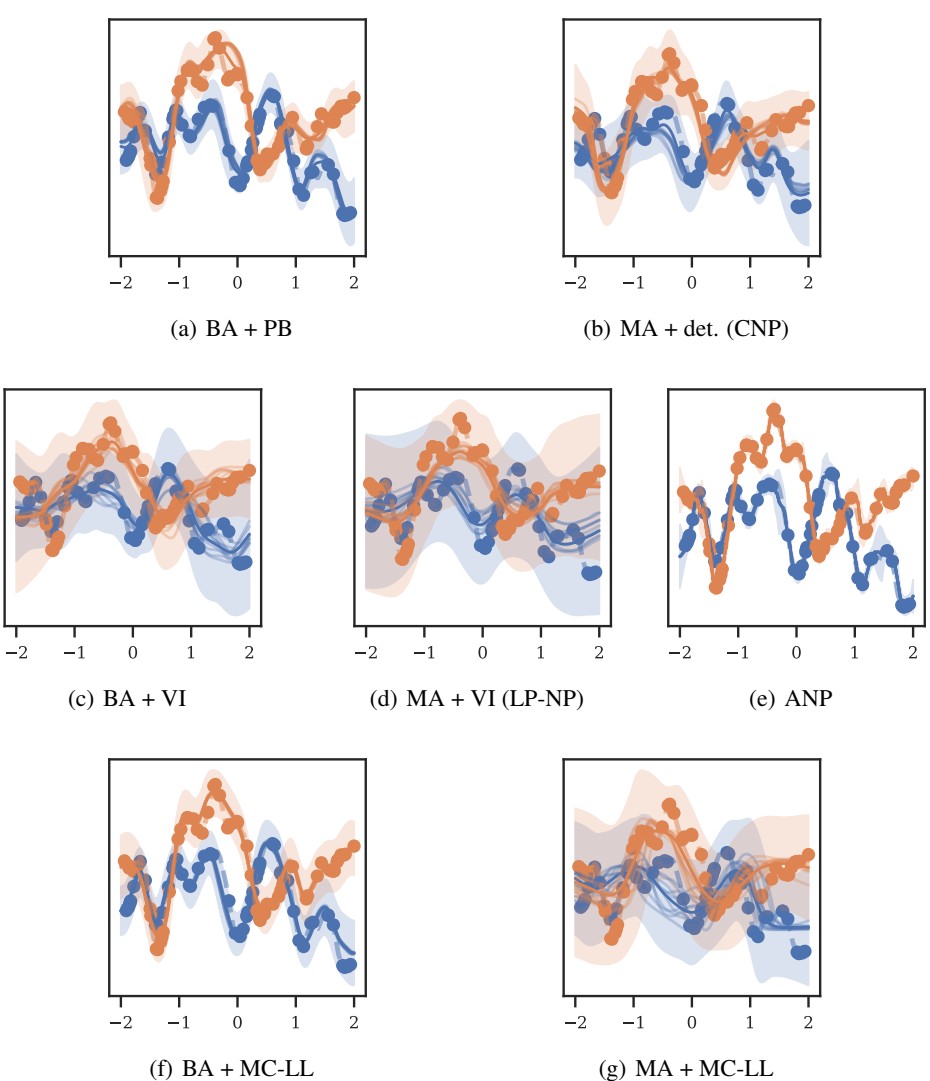

Figure 13: Predictions on two instances (dashed lines) of the Matern-5/2 GP function class, given $N = 60$ context data points (circles). We plot mean and standard deviation (solid line, shaded area) predictions together with 10 function samples (for deterministic methods we employ AR sampling).

