# OpenReview forum: "Bayesian Context Aggregation for Neural Processes"
_ICLR.cc/2021/Conference — ICLR 2021 Poster_

### Official Review · AnonReviewer4 · 2020-10-24
**A simple and effective idea, but missing an important baseline that can also address the problems of mean-aggregation in NPs**

**Rating:** 6
**Confidence:** 5

**Review:**

The authors present the Bayesian Aggregation (BA) mechanism in the context of Neural Processes (NPs) for aggregating the context information into the latent variable z in the form of posterior updates to z. The authors show that this improves predictive performance (in terms of likelihood) compared to mean aggregation MA that it replaces on various regression tasks with varying input-output dimensionality.

Strengths:
1. The idea is simple and leads to a notable improvement compared to MA in terms of likelihood
2. The background and method is presented very clearly.
3. The evaluation is done on a wide variety of tasks, ranging from standard 1D regression of GP samples to pendulum trajectory prediction tasks.

Weaknesses:
1. The evaluation is missing an important baseline model, which are (A)NP models that have self-attention in the encoder for processing the contexts (c.f. model figure in ANP paper (Kim et al., 2019b)). Contrary to the NP/CNP baselines that are compared against in the paper, the ANP with self-attention in the encoder does not give uniform weights to each context point - the self-attention allows the model to assign varying importance to the different context points (despite using mean-aggregation after the self-attention), which is presented as a key motivation for the BA mechanism introduced in the paper. Hence for the experiments, I strongly suggest comparing against CNP/NP/ANP with self-attention in the deterministic/latent/latent path of the encoder. For completeness, if would be nice to also compare against models that have both deterministic and latent paths, since BA can also be applied to these models. At the same time, I understand that BA would be more interpretable for showing which observations have little/high effect on z compared to the approach of using self-attention in the encoder, but it would still be very informative for the reader to be able to compare the two approaches. Also these two approaches can be combined to have self-attention in the encoder + BA, which might also yield improved performance.
2. The claim that “BA includes MA as a special case” doesn’t seem to be true. Using a “non-informative prior and uniform observation variances” leads to constant sigma_z and mu_z being linearly proportional to mean(r_n) (i.e. sum_n r_n / N), which is not quite the same as MA - MA allows sigma_z and mu_z to be non-linear functions of mean(r_n), hence is strictly more expressive than this special case.
3. In Equation (7), it seems as though the context points (x_n,y_n) only affects r_n via the variance, which seems unnecessarily limiting. Why not have the mean also depend on r_n? e.g. p(r_n|z) = N(r_n| z + mu_{r_n}, diag(sigma_{r_n}^2) where mu_{r_n} is also computed as a function of (x_n,y_n)? This will still give a closed-form posterior p(z|r_{1:N}) since the mean of p(r_n|z) is still linear in z, creating a model that’s strictly more expressive with very similar efficiency. It would be informative to see how this changes the experimental results.
4. I’m guessing the VI objective was used to train the ANP. Given the clear advantage of training with the MC objective, shouldn’t the ANP also be trained with MC?
5. The latent variable models were not evaluated on 2D image completion tasks because “architectures without deterministic paths were not able to solve this task”. Why not then add a deterministic path to these latent variable models to allow them to train?

Other points
- In the text, it says that the model is also compared against ANP to show that BA can compete with SOTA. This is arguably incorrect since ConvCNP models are SOTA among models of the NP family, showing a significant improvement over ANP. Hence to achieve the goal mentioned in the text, it would make sense to compare with ConvCNP models as part of the evaluation against other deterministic NPs.

Overall the paper is presented very clearly with a simple yet effective idea tested on a wide variety of tasks. However it’s missing an important baseline that uses self-attention in the encoder, along with several other baselines that would be informative to compare against. I am willing to increase my score should these results be included in the revised version of the paper.

=================

Score raised to 6 after inclusion of MA + SA results in rebuttal.

---

> ### Author Response · Authors · 2020-11-20
> **Author answer (part 2)**
>
> ***Please also consider the first part of our answer above***
>
> 2.)	MA as a special case of BA:
> As you pointed out correctly, in the discussion on p. 5 of our initial paper version, we claim that Eq. (8) reduces to the mean-aggregated latent observation $\bar{r}$ as given by Eq. (6) for non-informative priors and uniform observation variances. We believe that this claim is correct. Note that we explicitly refer to the mean-aggregated latent observation $\bar{r}$ **before** non-linear mappings as given by the $r$-to-$z$ networks. Thank you for pointing out that our discussion was potentially misleading in this respect. We clarify this in the revised version.
>
> 3.)	Extending the BA-observation model:
> We agree that your proposed observation model $p(r_n|z) = N(r_n | z + \mu_{r_n}, \mathrm{diag}(\sigma_{r_n}^2))$ with $\mu_{r_n}$ being a third encoder output (in addition to $r_n$ and $\sigma_{r_n}^2$ as proposed in our paper) would still allow for a closed-form posterior $p(z | r_{1:N})$. Indeed, in comparison to Eq. 8, $\sigma_z^2$ stays unchanged and $\mu_z$ now reads: $\mu_z = \mu_{z,0} + \sigma_z^2 \odot \sum_{n=1}^N (r_n - \mu_{r_n} - \mu_{z,0}) \oslash \sigma_{r_n}^2$. Note that $\mu_{r_n}$ enters this equation only subtracted from $r_n$. Therefore, this observation model does **not** represent a more expressive model than our proposed version without $\mu_{r_n}$, as we would just add up two distinct encoder outputs computed from the same inputs, which does **not** increase expressivity. This is why we do not consider this possibility in our paper.
>
> 4.)	Training loss for ANPs:
> You are right in that ANPs are defined in [1] to be trained with VI. Also, the reference implementation [2] we used considers only VI, which is why we only provide results for ANP+VI. As we mention in Sec. 6, our experiments show that the MC loss is a promising candidate to replace VI in future work on NPs.
>
> 5.)	NPs with parallel latent and deterministic paths:
> We agree that using BA in NP-based models with parallel latent and deterministic paths is an interesting direction for future work. However, it is outside of the scope of this paper as the primary focus of our work is to provide a clear and concise comparison of our proposed BA with traditional MA. Therefore, we designed our experiments with the goal to extract and assess the influence of the aggregation mechanism itself, without the influence of any other confounding factors. This is why we made sure to consistently compare architectures of the same type and perform extensive hyperparameter optimization, individually for each architecture and individually for each experiment to ensure a fair comparison. We believe that considering more complex architectures (such as architectures containing parallel deterministic and latent paths) would not add too much informative value to this intended focus of our experiments, in particular because there are numerous different design choices to consider here (e.g.: does the encoder attached to the deterministic path receive mean-aggregated quantities $\bar{r}$ or Bayesian-aggregated quantities $\mu_z$, $\sigma_z$, or both? How to properly combine cross-attention with BA? ...). Nevertheless, as stated in Sec. 6, we agree that all these novel architectural options are very interesting and promising approaches which should definitely be investigated in future work.
>
> 6.)	ConvCNPs as SOTA:
> Thank you for pointing this out. We add a remark about ConvCNPs in the revised paper version.
>
> We hope that the addition of your proposed SA-baseline together with the extended discussion will convince you that our paper is a valuable addition to the NP literature and ready for publication at ICLR. If so, we would be grateful if you reconsidered your score.
>
> [1] Kim et al., "Attentive Neural Processes", ICLR 2019
>
> [2] https://github.com/deepmind/neural-processes/blob/master/attentive_neural_process.ipynb
>
> [3] Vaswani et al., "Attention is all you need", NeurIPS 2017

---

> > ### Comment · AnonReviewer4 · 2020-11-23
> > **Reviewer response to author rebuttal**
> >
> > 1. Thank you for providing the comparison with self-attentive encoders. I see that although BA performs simlarly to MA + SA, it does offer some computational advantages. I'm a little sceptical of how beneficial these advantages are, given that the number of context points used for training is usually quite small. However I'm happy with the inclusion of this important baseline, hence will raise my score to 6.
> > 2. I see how Eq (8) can be reduced to (6), but if there are non-linear mappings in the r-to-z networks, then won't the resulting q(z|context) distribution in the MA model will be different to Eq (8)? I can't see how Eq (8) can cover this case. If this is correct, then I think it should be pointed out that BA cannot generalise MA when there exist non-linear mappings in the r-to-z network (which is usually the case).
> > 3. Thank you for the explanation. I think it is worth including this point in the paper for readers who might pose the same question.
> > 4. Can you not also train ANP with MC loss and add the results to the paper for completeness? It is relevant for this work because one of the things you show is that MC estimation is clearly superior to VI. I imagine the experiments will be simple to run given the current codebase.
> > 5. If adding results for latent-variable models on the image completion task is a stretch, I think it would be informative to at least add MA + SA results for the camera-ready version.

---

> > > ### Author Response · Authors · 2020-11-23
> > > **Author answer**
> > >
> > > Dear AnonReviewer4,
> > >
> > > 1. Thank you for your fair assessment and for raising your score!
> > > 2. It is correct that BA does not generalize MA if r-to-z mappings are included. Our argumentation considers $\bar{r}$ as the aggregated quantity for MA. In contrast, we motivate BA as a method which aggregates the context data directly in the statistical description of $z$, i.e., here we consider the distribution of z, described by $\mu_z$ and $\sigma_z^2$ to be the aggregated quantity. Comparing the aggregated quantities in this sense, our claim is correct. We will update the discussion again in the final revision in order to avoid confusion. Thanks again for insisting on clarity here!
> > > 3. Thank you for pointing this out! We will add a remark to clarify this in the final paper revision.
> > > 4./5. We agree that the results for ANP + MC as well as MA + SA on the image completion task would be informative. Unfortunately, we won't be able to present these results until the rebuttal deadline due to lack of time. Nevertheless, as you suggested, we will consider implementing ANP + MC as well as including the results for the image completion experiments in the camera-ready version.
> > >
> > > Thank you again for your thorough and constructive review!

---

> ### Author Response · Authors · 2020-11-20
> **Author answer (part 1)**
>
> Thank you for your detailed review, your positive remarks about our Bayesian aggregation (BA) mechanism, "a simple [idea which] leads to a notable improvement [...] on a wide variety of tasks", and for many insightful comments. We provide detailed answers for each of your comments below and revise our paper submission accordingly. In particular, we add comparisons with your proposed baseline employing self-attentive encoders, cf. Sec. 5 and App. 7.3.
>
> 1.)	Self-attentive encoders:
>
> We agree that an architecture with a self-attention (SA) mechanism in the encoder is an interesting baseline. Indeed, as you pointed out, SA also yields (similar to BA) a weighted sum of the latent observations $r_n$ (with the concrete definition of the weights depending on the type of SA mechanism used). In the initial version of our paper, we did not consider SA, because the reference implementation [2] of Attentive Neural Processes (ANPs) [1] (which first proposed to combine attention mechansims with Neural Processes (NPs)) does not include the option of SA in the encoder network.
>
> Inspired by your remarks, we re-implemented SA and evaluated three different versions (“Laplace”, “dot-product”, and “mulithead” self-attention, as proposed in [1,3]) in combination with mean aggregation (MA). For a fair comparison, we used the same extensive hyperparameter optimization procedure as for the other experiments. The results in Tabs. 6,7 in App. 7.3 show that SA can improve the performance of NP-based models drastically in comparison to traditional architectures employing MA without SA in the encoder. Nevertheless, our BA still performs better or at least on-par **without using SA**.
>
> We further add a detailed discussion of architectural and computational aspects of SA + MA in comparison with BA as well as considerations about how to combine BA with SA to the revised paper version. SA weights each latent observation according to some form of spatial relationship of the corresponding input with all other latent observations in the context set. In contrast, BA’s weight for a given latent observation is based only on features computed from this very latent observation and allows to incorporate an estimation of the amount of information contained in each context tuple into the aggregation. This leads to several computational advantages of BA over SA:
>
> (i) in general, SA scales quadratically in the number $N$ of context tuples, as it has to be evaluated on all $N^2$ pairs of context tuples. In contrast, BA scales linearly with $N$,
>
> (ii) BA allows for efficient incremental updates when context data arrives sequentially, while using SA does not provide this possibility: it requires to store and encode the whole context set at once and to subsequently aggregate the whole set of resulting (SA-weighted) latent observations,
>
> (iii) as you pointed out, combining BA with SA indeed sounds like an interesting and promising avenue of research. Note that BA relies on a second encoder output $\sigma_{r_n}^2$ (in addition to the latent observation $r_n$) which assesses the information content in each context tuple $(x_n, y_n)$. As each SA-weighted $r_n$ is informed by the other latent observations in the context set, obviously, one would have to also process the set of $\sigma_{r_n}^2$ with some form of SA mechanism. We are confident that this is indeed possible in a theoretically well-founded manner, but note that it is not immediately obvious how to do this properly. Therefore, we leave a combination of SA and BA for future research.
>
> ***Please also consider the second part of our answer below***

---

### Official Review · AnonReviewer1 · 2020-10-26
**An interesting Paper Proposing a Sound solution**

**Rating:** 7
**Confidence:** 3

**Review:**

Summary of the Paper:

        This paper describes Bayesian context aggregation for neural processes. These models are useful to address regression problems in which a set of related tasks are available for inference with associated context information in the form of extra data. These models assume that there is a task-specific global latent variable and task-independent latent variable. They are learned via approximate maximum posterior likelihood, in which the latent variables specific for each tasks are marginalzied out. For this, an approximation to the posterior distribution of these variables is need. This requires conditioning to the context dataset which is challenging. In the past, a latent representation is used and the context data set is aggregated as the mean of the latent representation. In this paper a Bayesian way of aggregating context information is proposed. This is based on using Bayes rule and a Gaussian generative model for the latent representations. The proposed method also leads to a new way of training CLV models which is based on moment matching. The method is validated on several synthetic an real-world experiments showing improvements over mean aggregation.

Detailed Comments:

I believe that this is a relevant paper. Context aggregation is a difficult problem that is required to address the learning tasks described in the paper. Previous solution look limited and the proposed method seems natural and a more effective method of aggregating this information. The paper is well written and the proposed method is sound. The experiments are also convincing and exhaustive. I believe that this is a relevant paper for the conference.

---

> ### Author Response · Authors · 2020-11-20
> **Author answer**
>
> Thank you very much for your review and your positive remarks, judging our paper to be "relevant for the conference", as it proposes Bayesian context aggregation, a "sound solution" for a "difficult problem".

---

### Official Review · AnonReviewer3 · 2020-10-28
**A solid improvement for neural process inference, but I'd like to see actual regression tasks**

**Rating:** 6
**Confidence:** 4

**Review:**

In this paper, the authors make two contributions to neural process-like CLV models. First, they replace the somewhat adhoc variational-like approach to learning the amortized latent variable distribution with a monte carlo based approximation. Second, they replace the step of context aggregation with direct latent variable inference over z.

Overall, in my opinion these modifications make the neural process model significantly cleaner from a Bayesian perspective and is quite nice. In the context of neural processes, I have very little criticism for the authors' methods. Everything makes sense, and the MC approach in equation (3) seems cleaner to me than the somewhat ad hoc "VI like" approach in equation (2).

The biggest difficulty I have is determining how to evaluate the authors' clear improvements to neural processes in the broader context of scalable probabilistic regression, an area to which the authors claim membership. To start with, the authors clearly demonstrate the value of both Bayesian context aggregation and a MC based likelihood approximation scheme on precisely the same types of problems that existing neural processes papers (e.g., Garnelo et al., 2018) have considered (with the notable exception that the 2D image completion task considers only MNIST as a target dataset). In this respect, it's difficult to fault the experimental evaluation.

However, this paper and many neural process papers are written in the context of "Formulating scalable probabilistic regression models with reliable uncertainty estimates." Surely, at some point, this should involve a comparison of these approaches to existing techniques for probabilistic regression, whether that be deep Gaussian processes, dropout based approaches, Bayesian neural networks, or other approaches. I don't mean to imply here that the authors are unaware of this large body of literature -- indeed, the authors have a decent if incomplete overview of techniques in this area (notably missing work on deep GPs).

Rather, it just seems surprising to me that the discussion of the relevant probabilistic regression literature ends at "well, it exists." What I would like to see is a discussion of where the authors' impressive improvements to neural processes leave the model family in this broader context. How close or far off is the family on performance for standard benchmark regression tasks? Are there settings in which we can leverage the fully NN based nature of neural processes to achieve probabilistic regression in settings where the inductive biases of kernel methods are poor, like in computer vision or natural language processing? The relatively toy nature and limited dimensionality of the problems considered suggests that there is still significant progress to be made before such a comparison would be reasonable or even possible.

To summarize, in the context of neural processes I feel the paper makes good methodological contributions in presenting a much cleaner and more natural (from a Bayesian perspective) version of the model that has more of the flavor of standard amortized inference for latent variable models. Within the very narrow context of neural process papers, I therefore have very little to complain about. However, from a broader scientific perspective I would feel that the paper would be significantly strengthened by a fair evaluation to the rest of this literature, whether empirical or simply in discussion, regardless of how the authors' approach fares in comparison.

---

> ### Author Response · Authors · 2020-11-20
> **Author answer (part 2)**
>
> ***Please also consider the first part of our answer above***
>
> As you pointed out correctly, NPs are motivated in [1-3] as models which combine the computational efficiency of neural networks with well-calibrated uncertainty estimates (like those of GPs). Indeed, NPs scale linearly in the number N of context and M of target data points, i.e., like $\mathcal O(N+M)$ while GPs scale like $\mathcal O(N^3 + M^2)$. Furthermore, NPs are shown to exhibit well-calibrated uncertainty estimates. In this sense, NPs can be counted as members of the family of scalable probabilistic regression methods.
>
> However, we might not have emphasized clearly enough the following central aspect of NP training which is relevant to clarify your question. We apologize for that and improve our exposition in the revised paper version. NPs are trained in a multi-task fashion (note that this is true not only for our contribution, but holds in general for NP-based model architectures). This means that NPs rely on data from a set of related source tasks from which they automatically learn powerful priors and the ability to adapt quickly to unseen target tasks. This multi-task training procedure of NPs scales linearly in the number $L$ of source tasks, which makes it possible to train these architectures on large amounts of source data. Applying GPs in such a multi-task setting can be challenging, especially for large numbers of source tasks [4-6]. Similarly, Bayesian Neural Networks (BNNs) [7,8] as well as DeepGPs [9] are in their vanilla forms specifically designed for the single-task setting. Therefore, GPs, BNNs, and DeepGPs are not directly applicable in the NP multi-task setting, which is why they are typically not considered as baselines for NP-based models, as discussed in [3].
>
> The experiments presented in [1-3] focus mainly on evaluating NPs in the context of few-shot probabilistic regression, i.e., on demonstrating the data-efficiency of NPs on the target task after training on data from a range of source tasks. In contrast, the application of NPs in situations with large ($>1000$) numbers of context/target points per task has to the best of our knowledge not yet been investigated in detail in the literature. Furthermore, it has not been studied how to apply NPs in situations where only a single or very few source tasks are available. The focus of our paper is a clear-cut comparison of the performance of our BA with traditional mean aggregation (MA) in the context of NP-based models. Therefore, we also consider experiments similar to those presented in [1-3] and leave further comparisons with existing methods for (multi-task) probabilistic regressions for future work.
>
> Nevertheless, to illustrate this discussion, we add two simple GP-based baseline methods for this rebuttal:
>
> (i) a vanilla GP, i.e., the hyperparameters are optimized on each target task individually and the source data is not used, and
>
> (ii) a naive but easily interpretable example of a multi-task GP which optimizes one set of hyperparameters on all source tasks and uses it for predictions on the target tasks without further adaptation.
>
> The results in Tab. 8 in App. 7.4 show that those simple GP-based models can only compete with NPs on function classes where either the inductive bias as given by the kernel functions fits the data well, or on function classes which exhibit a relatively low degree of variablity. On more complex function classes NPs produce predictions of much better quality, as they incorporate the source data more efficiently.
>
> We further add missing literature about DeepGPs [9] and about more elaborate versions of multi-task GP regression methods [4-6] to the revised version of our paper.
>
> We hope that this discussion together with the illustrative GP-baselines can help to clarify your question.
>
>
>
> [1] Garnelo et al., "Conditional Neural Processes", ICML 2018
>
> [2] Garnelo et al., "Neural Processes", ICML 2018 workshop on Theoretical Foundations and Applications of Deep Generative Models
>
> [3] Kim et al., "Attentive Neural Processes", ICLR 2019
>
> [4] Bardenet et al., "Collaborative Hyperparameter Tuning", ICML 2013
>
> [5] Yogatama and Mann, "Efficient Transfer Learning Method for Automatic Hyperparameter Tuning", AISTATS 2014
>
> [6] Golovin et al., "Google Vizier: A Service for Black-Box Optimization", International Conference on Knowledge Discovery and Data Mining, 2017
>
> [7] MacKay, "A Practical Bayesian Framework for Backpropagation Networks", Neural Comput., 1992
>
> [8] Gal and Ghahramani, "Dropout as a Bayesian Approximation: Representing Model Uncertainty in Deep Learning", ICML, 2016
>
> [9] Damianou and Lawrence, "Deep Gaussian Processes", ICML 2013

---

> ### Author Response · Authors · 2020-11-20
> **Author answer (part 1)**
>
> Thank you very much for your detailed comments and the positive review of our paper, which rates our novel Bayesian aggregation (BA) mechanism and our evaluation of MC-based likelihood approximations as a "solid improvement for neural process inference" which "make the neural process model significantly cleaner from a Bayesian perspective".
>
> We gladly address your remaining question about where Neural Process (NP)-based models are located on the map of (scalable) probabilistic regression methods by adding a detailed discussion together with an illustrative comparison with GP-based baselines to the revised version of our paper, cf. App. 7.4.
>
> ***Please also consider the second part of our answer below***

---

### Official Review · AnonReviewer2 · 2020-11-01
**A new Bayesian regression model as a multi-task learning problem**

**Rating:** 6
**Confidence:** 1

**Review:**

The paper builds upon previous lines of research on multi-task learning problem, such as conditional latent variable models including the Neural Process. As shown by the extensive Related Work section, this seems to be an active research direction. This makes it difficult for me to judge originality and significance, but it is well-written and clear.

Specific comments
- the approximate posterior distribution q_\phi is often referred to as "the posterior distribution". I would keep "approximate" here.
- p2: "correspondence of GPs with infinite Bayesian NNs (BNNs)", what is meant by "infinite BNN"? is it infinite-width BNN? please specify.
- p2: "adaptive BLR" please describe the acronym.
- p6: the Gaussian approximation of the posterior predictive likelihood (10) is said to be "inspired by GPs which also define a Gaussian likelihood". This is also essentially what is done by Synthetic Likelihood (a Nature paper by SN Wood, 2010) which is I think more closely related to the proposed approach than GPs.
- p6, one line below: define PB acronym the first time it is used (not three lines after).

Typos
- top p2: "does not introducing noticeable computational overhead".
- p6: "conditional model with an decoder operating on".
- The references list could be tidied: some first names are abbreviated, some not.

---

> ### Author Response · Authors · 2020-11-20
> **Author answer**
>
> Thank you very much for your positive review of our paper and for your comments, which we address in the revised paper version!
>
> We would further like to comment on your remark about originality and significance of our work. Neural Processes (NPs) are a well-established model family for probabilistic few-shot regression, proposed in [1-3]. A central component of such models is a context data aggregation which constitutes "a difficult problem" (AnonReviewer1). Traditionally, NP-based models use mean-aggregation (MA), a solution which "look[s] limited" (AnonReviewer1). Our contribution is Bayesian aggregation (BA), a novel, "much cleaner and more natural (from a Bayesian perspective)" (AnonReviewer3), and "simple and effective" (AnonReviewer4) aggregation mechanism. We can show that it is compatible with existing NP-based architectures and "leads to notable improvement compared to MA in terms of likelihood" (AnonReviewer4). Therefore, we are convinced that our paper constitutes an original and significant contribution to the literature.
>
> [1] Garnelo et al., "Conditional Neural Processes", ICML 2018
>
> [2] Garnelo et al., "Neural Processes", ICML 2018 workshop on Theoretical Foundations and Applications of Deep Generative Models
>
> [3] Kim et al., "Attentive Neural Processes", ICLR 2019

---

### Author Response · Authors · 2020-11-20
**Paper Revision**

Dear reviewers!

Thank you very much for your generally positive reviews of our paper and for numerous constructive remarks! We uploaded a revised version of the paper where we highlighted updated passages in blue and added two new paragraphs to the appendix (App. 7.3, 7.4). We further provide detailed comments on each of your reviews below.

Kind regards,
The authors

___________________________________________


***Updates***

Nov 24th, 15:08pm (GMT+1): We uploaded a new version of the paper revision, including some further clarifications according to AnonReviewer4's remarks.

Nov 23rd, 11:57am (GMT+1): We uploaded a new version of the paper revision, where we put the results of the new baseline employing self-attentive encoders as proposed by AnonReviewer4  into the main part of the text, cf. end of Sec. 5.

Nov 22nd, 6:00pm (GMT+1): We uploaded a new version of the paper revision where we improved some passages of the discussion and added further experimental results to App. 7.3.

---

### Decision · Program_Chairs · 2021-01-07
**Final Decision**

**Decision:**

Accept (Poster)

**Comment:**

The authors present a Bayesian approach for context aggregation in neural processes based models. The article is well written, and provides a nice and comprehensive framework. The reviewers raised some issues regarding the lack of comparisons to proper baselines. The authors provided additional comparisons in the revised version. The comparisons were found satisfactory by some some reviewers, who increased their scores. Based on the revised version, I recommend acceptance.